# Stroke subtype-dependent synapse elimination by reactive gliosis in mice

Xiaojing Shi[1,6], Longlong Luo[1,5,6], Jixian Wang[2,6], Hui Shen[1], Yongfang Li[2], Muyassar Mamtilahun[1], Chang Liu[1], Rubing Shi[1], Joon-Hyuk Lee [3], Hengli Tian[1], Zhijun Zhang[1], Yongting Wang[1], Won-Suk Chung [3✉], Yaohui Tang [1✉] & Guo-Yuan Yang [1,4✉]

The pathological role of reactive gliosis in CNS repair remains controversial. In this study, using murine ischemic and hemorrhagic stroke models, we demonstrated that microglia/macrophages and astrocytes are differentially involved in engulfing synapses in the reactive gliosis region. By specifically deleting MEGF10 and MERTK phagocytic receptors, we determined that inhibiting phagocytosis of microglia/macrophages or astrocytes in ischemic stroke improved neurobehavioral outcomes and attenuated brain damage. In hemorrhagic stroke, inhibiting phagocytosis of microglia/macrophages but not astrocytes improved neurobehavioral outcomes. Single-cell RNA sequencing revealed that phagocytosis related biological processes and pathways were downregulated in astrocytes of the hemorrhagic brain compared to the ischemic brain. Together, these findings suggest that reactive microgliosis and astrogliosis play individual roles in mediating synapse engulfment in pathologically distinct murine stroke models and preventing this process could rescue synapse loss.

[1] School of Biomedical Engineering and Affiliated Sixth People's Hospital, Shanghai Jiao Tong University, Shanghai 200030, China. [2] Department of Rehabilitation, Ruijin Hospital, School of Medicine, Shanghai Jiao Tong University, Shanghai 200025, China. [3] Department of Biological Sciences, Korea Advanced Institute of Science and Technology, Daejeon 34141, South Korea. [4] Department of Neurology, Ruijin Hospital, School of Medicine, Shanghai Jiao Tong University, Shanghai 200025, China. [5] Present address: Dermatology and Venerology Unit, Department of Medicine, Karolinska Institute, Stockholm, Sweden. [6] These authors contributed equally: Xiaojing Shi, Longlong Luo, Jixian Wang. ✉email: wonsuk.chung@kaist.ac.kr; yaohuitang@sjtu.edu.cn; gyyang@sjtu.edu.cn

Both ischemic and hemorrhagic stroke are the leading causes of death and disability worldwide[1]. At the recovery stage of ischemic and hemorrhagic stroke, microglia/macrophages, reactive astrocytes, and NG2 glia highly proliferate and start to form reactive gliosis, eventually forming glial scars in the lesion area[2]. The glial scar is traditionally considered a physical and chemical barrier to central nervous system (CNS) regeneration, as it forms dense isolation and creates an inhibitory environment for axon regeneration and remyelination[3,4]. However, accumulating evidence has demonstrated that glial scar formation also aides CNS axon regeneration[5–7]. A better understanding of the physiological function of reactive gliosis is critical for developing precise therapy.

Microglia and astrocytes are highly responsive resident brain cells that dramatically change their features in response to stroke[8,9]. Transcriptomic profiling showed that both microglia and astrocytes were enriched in synapse engulfment pathway-related genes, including phagocytic receptors, intracellular molecules, and opsonins[10]. Astrocytes use the MEGF10 and MERTK phagocytic pathways while microglia/macrophages use the classical complement pathway to mediate synapse elimination in developing[11,12], and diseased brains[13,14]. In stroke brain, dead neurons and synaptic debris in microglia, macrophages, and astrocytes were observed[15,16]. Although microglia/macrophage and astrocyte-mediated phagocytosis are generally assumed to be necessary for clearing synaptic debris and beneficial for brain recovery, recent studies have suggested that phagocytic microglia/macrophages and astrocytes also damage live synapses in Alzheimer's disease[17,18] and viable neurons in stroke[19]. Microglia/macrophages phagocytose viable neurons through MFG-E8 and MERTK at the acute stage of ischemic stroke, and inhibiting this process prevents delayed loss and death of functional neurons[19]. Although previous studies have highlighted the important roles of microglia/macrophages and astrocytes in neuronal debris clearance during the early stage of stroke, whether they are still phagocytic and cause synapse loss at the subacute and recovery stages remains largely unknown.

In the present study, we identified a role for reactive microgliosis and astrogliosis in engulfing synapses in the poststroke repair and remodeling stage. Compared with the previous view that glial phagocytosis has a protective role in stroke, we demonstrated that reactive microgliosis and astrogliosis hinder brain repair by engulfing synapses, and specific inhibition of this phagocytosis before glial scar maturation has beneficial consequences, including reducing synapse loss and improving neurobehavioral outcomes. More importantly, we further showed that reactive microgliosis- and astrogliosis-mediated phagocytosis are stroke subtype-dependent, providing therapeutic targets and potential treatment strategies in different stroke subtypes.

## Results

**Phagocytosis of synapses by microglia/macrophages and astrocytes in ischemic and hemorrhagic stroke mice.** Experiments were designed as shown in Supp. Fig. 1. To determine whether reactive microglia/macrophages and astrocytes become phagocytic after stroke, the phagocytic biomarker Mac-2[20], was costained with the microglia/macrophage marker Iba-1 and astrocyte marker GFAP at 1, 3, 7, and 14 days after stroke. In ischemic stroke, Mac-2 was found in both microglia/macrophages and astrocytes (Fig. 1a). Interestingly, Mac-2 expression in microglia/macrophages peaked at day 3 after middle cerebral artery occlusion (MCAO) and then gradually reduced at day 7 and day 14, whereas Mac-2 expression in astrocytes was elevated from day 1 to day 14 (Fig. 1a). Similarly, Mac-2 was also colocalized with microglia/macrophages and astrocytes in the

hemorrhagic mouse brain (Fig. 1g). By quantifying the number of Mac-2$^+$/Iba-1$^+$ microglia/macrophages and Mac-2$^+$/GFAP$^+$ astrocytes in the gliosis region at 14 days after MCAO, we found that 25% of microglia/macrophages and 50% of astrocytes expressed Mac-2 (Fig. 1e). Three days after hemorrhagic stroke, >50% of microglia/macrophages and only 20% of astrocytes expressed Mac-2. The number was gradually reduced at day 7 and day 14, with 25% of microglia/macrophages and only 5% of astrocytes expressing Mac-2 (Fig. 1k).

Following stroke, reactive microglia/macrophages and astrocytes become highly proliferative and form gliosis regions to isolate the necrotic area from the rest of the brain (Fig. 1b, h). We then investigated whether activated microglia/macrophages and astrocytes could phagocytose synapses in ischemic and hemorrhagic mouse brains. We found that 14 days after ischemic stroke, astrocytes and microglia/macrophages engulfed synaptic elements immunolabeled for both presynaptic (synaptophysin, SYP) and postsynaptic (Homer-1) proteins in the microgliosis and astrogliosis areas (Fig. 1b). Confocal imaging demonstrated that SYP$^+$ presynaptic protein and Homer-1$^+$ postsynaptic protein were detected inside microglia/macrophages and astrocytes (Fig. 1b, f), indicating that entire synapses were engulfed. However, 14 days after hemorrhagic stroke, both SYP$^+$ and Homer-1$^+$ proteins were engulfed by microglia/macrophages but few were engulfed by astrocytes (Fig. 1h, l). To distinguish the engulfing contribution of resident microglia and infiltrated macrophages, we used the microglia-specific marker P2RY12[21] and macrophage-specific marker F4/80[22], colabeled with synaptic markers (Supp. Fig. 2). The number of phagocytic microglia (P2RY12$^+$/SYP$^+$ and P2RY12$^+$/Homer-1$^+$ cells) and phagocytic macrophages (F4/80$^+$/SYP$^+$ and F4/80$^+$/Homer-1$^+$ cells) was quantified. We found that microglia contributed more to engulfing synapses than macrophages in both ischemic and hemorrhagic stroke.

To further confirm that reactive microglia/macrophages and astrocytes completely engulfed synaptic materials, the gliosis area was imaged using transmission electron microscopy (TEM). Immunolabeled microglia/macrophages and astrocytes were recognized by their 3,3′-diaminobenzidine (DAB) electron-dense precipitates within the cytoplasm (Supp. Fig. 3). Consistent with the immunostaining results, TEM data showed reactive microglia/macrophages and astrocytes containing synaptic elements, which were located in the cytoplasm at 14 days after ischemic stroke (Fig. 1c). Statistical analysis showed that microglia/macrophages and astrocytes contribute similarly to engulfing synaptic vesicles (Fig. 1d). Magnified views of the engulfed synaptic structure displayed multivesicular and irregular membrane morphology (Fig. 1c), suggesting that the material was in the process of active degradation. However, synaptic elements were mainly detected in the cytostome of microglia/macrophages but rarely detected in astrocytes in the hemorrhagic mouse brain (Fig. 1i, j).

**Both inhibitory and excitatory synapses were engulfed by microglia/macrophages and astrocytes, and digested through lysosomal pathways.** To investigate which types of synapses were engulfed in the stroke mouse brain, we utilized recently developed synapse phagocytosis sensors[23]. Adeno-associated virus (AAV) carrying fluorescent phagocytosis reporters for inhibitory post-synapses (hsyn-Gephyrin-mCherry-eGFP) and excitatory post-synapses (hsyn-PSD95-mCherry-eGFP) was injected into the striatum region of the mouse brain. GFP (pKa 6.0) and RFP (pKa 4.5) show different sensitivities to the lysosomal environment. Lysosomal hydrolases degrade GFP first so that GFP fluorescence is attenuated in lysosomes, whereas RFP fluorescence is stable in

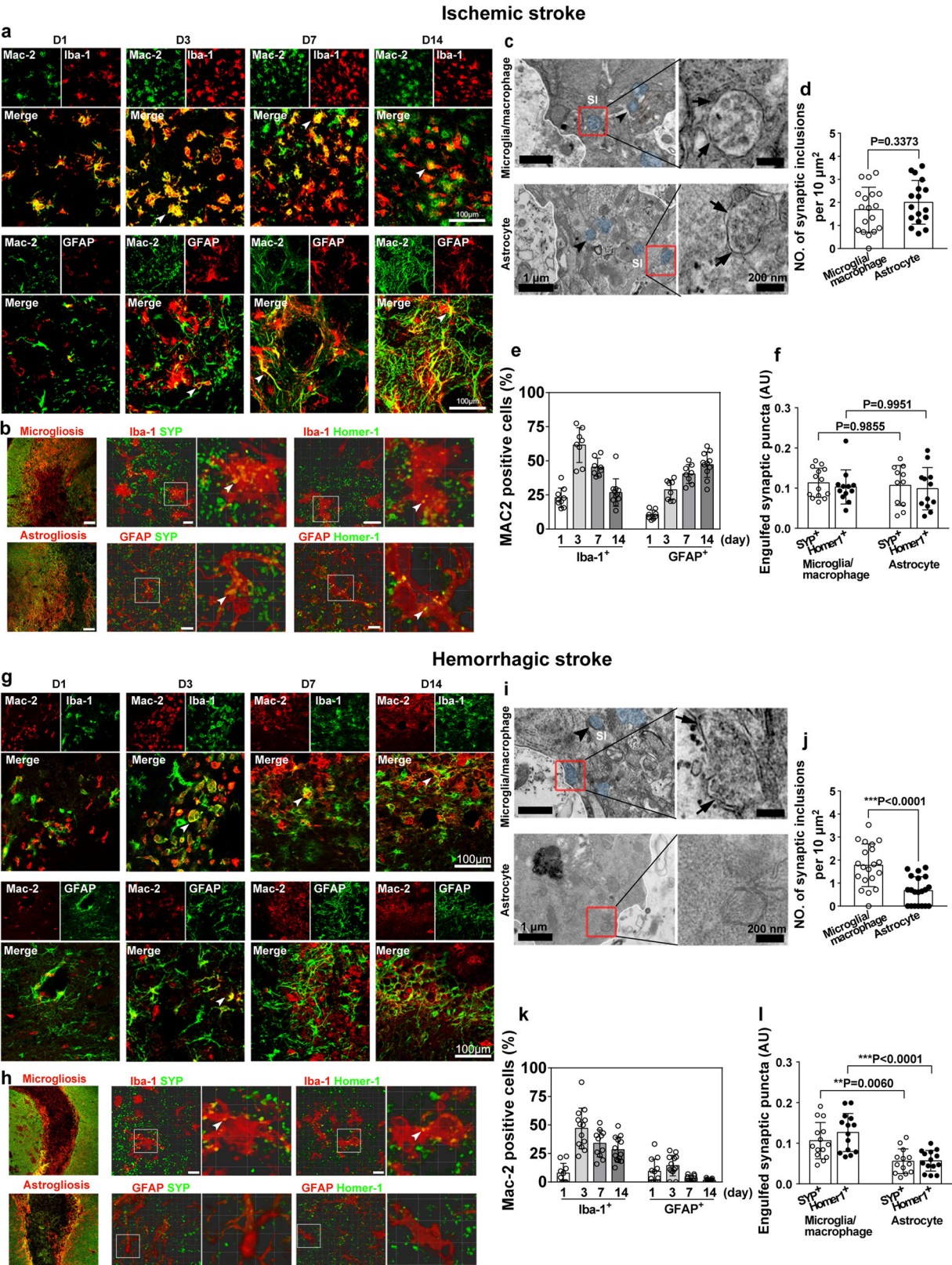

**Ischemic stroke**

**Hemorrhagic stroke**

lysosomes[24]. Based on this principle, we found that inhibitory postsynapses (InhiPost) and excitatory postsynapses (ExPost) were engulfed by both microglia/macrophages and astrocytes in the ischemic brain (Fig. 2a, d), and microglia/macrophages and astrocytes did not show a preference for phagocytosis of excitatory and inhibitory synapses, which might contribute to

protection against synaptic excitatory/inhibitory imbalance (Fig. 2b, e). Detailed analysis of synaptic puncta inside the glial process showed that in the ischemic brain, the cellular or extracellular localization of mCherry-GFP and mCherry-alone puncta was similar in both microglia/macrophages and astrocytes (Fig. 2c, f). In the hemorrhagic brain, most of InhiPost and

**Fig. 1 Engulfment of synapses by microglia/macrophages and astrocytes in the gliosis region in ischemic and hemorrhagic stroke.** Representative single-plane images showed immunostaining of Iba-1 (green), GFAP (green), and Mac-2 (red) in the perifocal region at 1, 3, 7, and 14 days after ischemic stroke (**a**) and hemorrhagic stroke (**g**). Arrowheads indicate Mac-2 expression in glial cells. Bar = 100 μm. Percentage of Iba-1$^+$/Mac-2$^+$ and GFAP$^+$/Mac-2$^+$ cells at different time points after ischemic (**e**) and hemorrhagic stroke (**k**). **b**, **f** Confocal images showed presynaptic protein synaptophysin (SYP$^+$, green) and postsynaptic protein Homer-1$^+$ (green) were engulfed by Iba-1$^+$ microglia/macrophages (red) and GFAP$^+$ astrocytes (red) in the reactive gliosis region at 14 days after MCAO. Left panel, low magnification of gliosis region in ischemic striatum; glial cell, red; synapse, green. Arrowheads indicate engulfed synapses. Bar = 100 μm and 10 μm. **c**, **d** TEM images showed engulfed synapses (blue) were contained in the cytoplasm of microglia/macrophages and astrocytes in the gliosis region following MCAO. High magnification images (right panel) from the box showed the engulfed synaptic structures with multivesicular and irregular membrane morphology (arrows) in microglia/macrophages and astrocytes. Bar = 1 μm and 200 nm. **h**, **l** Confocal images showed SYP$^+$ (green) and Homer-1$^+$ (green) were engulfed by Iba-1$^+$ microglia/macrophages (red) in the microgliosis area, but few SYP$^+$ (green) and Homer-1$^+$ (green) were detected in GFAP$^+$ astrocytes (red) in the astrogliosis area in the hemorrhagic brain. Left panel, low magnification of gliosis region in hemorrhagic striatum, glial cell, red; synapse, green. Arrowheads indicate engulfed synapses in microglia/macrophages. Bar = 100 μm and 5 μm. **i**, **j** TEM images showed engulfed synapses (blue) localized in the cytoplasm of microglia/macrophages but not in astrocytes following hemorrhagic stroke. High-magnification images (right panel) from the box (left panel) showed the synaptic inclusions (arrow) in microglia/macrophages but not in astrocytes. Bar = 1 μm and 200 nm. *SI* synaptic inclusion, *AU* arbitrary units. Statistics are derived from 18, 17 cells (**d**) and 20, 20 cells (**j**) (from left to right), n = 3 mice per group. Statistics are derived from 9, 8, 8, 11, 8, 8, 8, 11 slices (**e**) and 10, 13, 11, 13, 12, 13, 10, 11 slices (**k**) (from left to right), n = 4 mice per group. Statistics are derived from 14, 12, 11, 12 slices (**f**) and 14, 14, 13, 14 slices (**l**) (from left to right), n = 4 mice per group. **d**, **j** two-sided, unpaired Student's t test. **f**, **l** two-way ANOVA followed by Tukey's test. Data are mean ± SD. Source data are provided as a Source Data file.

ExPost were engulfed by microglia/macrophages (Fig. 2g, j), and microglia/macrophages and astrocytes did not show a preference for phagocytosis of excitatory and inhibitory synapses (Fig. 2h, k). For microglia/macrophages, approximately half of mCherry-alone puncta were inside glial cells, and <15% of mCherry-alone puncta were outside of glial cells (Fig. 2i); for astrocytes, there was over 30% of mCherry-alone puncta localized outside of glial cells (Fig. 2l), the proportion was much higher than that of microglia/macrophages, indicating astrocytes engulfed fewer synapses in the hemorrhagic brain.

In our study, we also notified GFP-negative mCherry puncta outside the labeled glial cells, which can originate from two different sources. First, as we labeled microglia/macrophages and astrocytes separately in Fig. 2, most mCherry puncta outside the labeled cells were likely located in nonlabeled microglia/macrophages or astrocytes. Second, in our previous publication[23], we also reported that a minor population of mCherry-alone puncta can be found inside neuronal processes, since synaptic vesicles can be internalized/recycled by neuronal organelles.

To confirm whether these glial cells digest engulfed synaptic materials by lysosomes, we stained microglia/macrophages and astrocytes for the lysosome marker LAMP2. We found that in the ischemic brain, over one-half of SYP$^+$ and Homer-1$^+$ signals were colocalized with LAMP2$^+$ signals in either microglia/macrophages (Fig. 3a, b) or astrocytes (Fig. 3c, d). Importantly, >60% of SYP$^+$ and Homer-1$^+$ signals were detected in the lysosomes of microglia/macrophages (Fig. 3e, f), whereas only ~25% of synapses were inside astrocyte lysosomes in the hemorrhagic brain (Fig. 3g, h). We also noted that some SYP$^+$ and Homer-1$^+$ signals did not colocalize with LAMP2$^+$ signals, suggesting that they are in phagosomal compartments prior to lysosomal degradation.

**Increased MEGF10 and MERTK expression in reactive gliosis regions in stroke mouse brain.** To determine whether MEGF10 and MERTK pathways mediate synapse engulfment after stroke, we investigated the expression and localization of MEGF10 and MERTK at different time points after stroke (Fig. 4a, k). We found that compared with the corresponding sham group, 14 days after stroke, MEGF10 and MERTK expression was significantly increased (Fig. 4b, l). Moreover, we found that MEGF10 and MERTK expression was highly upregulated in reactive microgliosis and astrogliosis, as shown by immunostaining (Fig. 4c, e, m, o). Fluorescence in situ hybridization (FISH) further showed that MEGF10 and MERTK were expressed in reactive gliosis in both ischemic (Fig. 4g–j) and hemorrhagic mouse brains (Fig. 4q–t). We also

examined MEGF10/MERTK expression in neurons (MAP2$^+$) and endothelial cells (CD31$^+$) (Supp. Fig. 4), and statistical analysis showed that MEGF10/MERTK was rarely expressed in MAP2$^+$ neurons or CD31$^+$ endothelial cells in the striatum in ischemic (Fig. 4d, f) or hemorrhagic brains (Fig. 4n, p).

**Conditional MEGF10 or MERTK knockout in microglia/macrophages or astrocytes reduced synapse engulfment in the gliosis region.** To further explore whether microglia/macrophages and astrocytes engulf synapses through MEGF10 and MERTK at the repair and remodeling stages after stroke, we analyzed the phagocytosis of microglia/macrophages and astrocytes by using astrocyte-specific MEGF10 knockout mice (Aldh1l1$^{Cre-ERT2}$; MEGF10$^{flox/flox}$), astrocyte-specific MERTK knockout mice (Aldh1l1$^{Cre-ERT2}$; MERTK$^{flox/flox}$), microglia/macrophage-specific MEGF10 knockout mice (CX3CR1$^{Cre-ERT2}$; MEGF10$^{flox/flox}$) and microglia/macrophage-specific MERTK knockout mice (CX3CR1$^{Cre-ERT2}$; MERTK$^{flox/flox}$). MEGF10$^{flox/flox}$ (MEGF10$^{WT}$) and MERTK$^{flox/flox}$ (MERTK$^{WT}$) mice were used as controls. We validated the MEGF10 and MERTK conditional knockout specificity by injecting AAV5-GFAP-Cre virus into MEGF10$^{flox/flox}$ or MERTK$^{flox/flox}$ brains. The data showed astrocyte-specific deletion of MEGF10 and MERTK proteins, compared with control virus injected cases (Supp. Fig. 5a). In addition, we validated the MEGF10 and MERTK conditional knockout specificity in the microglia/macrophage- or astrocyte-specific MEGF10 and MERTK KO groups with tamoxifen or oil injections, and the results exhibited a significant reduction in MEGF10 and MERTK expression, compared with the control groups (Supp. Fig. 5b). MEGF10 and MERTK protein level was significantly decreased in the conditional KO groups (Supp. Fig. 5b, c). Immunostaining further confirmed that MEGF10 and MERTK were specifically knocked out in microglia/macrophages but not in astrocytes in CX3CR1$^{Cre-ERT2}$; MEGF10$^{flox/flox}$ (C-MEGF10$^{KO}$) and CX3CR1$^{Cre-ERT2}$; MERTK$^{flox/flox}$ (C-MERTK$^{KO}$) mice (Supp. Fig. 5e, g). In Aldh1l1$^{Cre-ERT2}$; MEGF10$^{flox/flox}$ (A-MEGF10$^{KO}$) and Aldh1l1$^{Cre-ERT2}$; MERTK$^{flox/flox}$ (A-MERTK$^{KO}$) mice, we did not detect MEGF10 and MERTK in astrocytes, but MEGF10 and MERTK were colocalized with microglia/macrophages (Supp. Fig. 5f, h). We also validated that these gene manipulations did not affect the initial brain injury (Supp. Fig. 6) following acute ischemic and hemorrhagic stroke.

We then performed high-resolution confocal microscopy to analyze glial cell-mediated engulfment of the presynaptic protein SYP and the postsynaptic protein Homer-1 in the gliosis region at

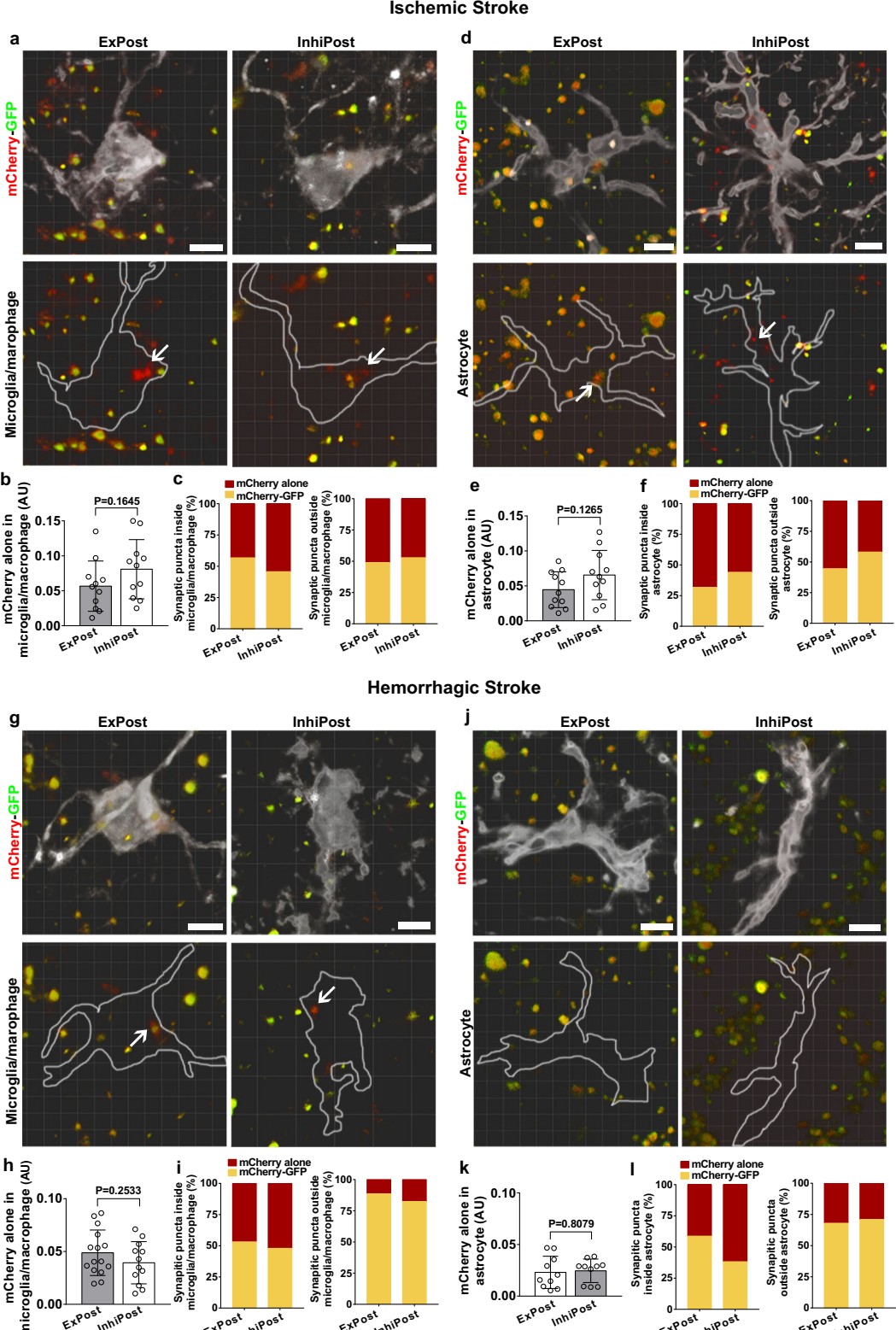

14 days after stroke. We found that 14 days after ischemic stroke, MEGF10$^{KO}$ or MERTK$^{KO}$ in microglia/macrophages or astrocytes resulted in reduced engulfment of SYP$^+$ and Homer-1$^+$ synaptic puncta in microglia/macrophage (Fig. 5a, b) or astrocytes (Fig. 5c, d) compared with controls. Consistent with our immunostaining data, western blotting of SYP and Homer-1 in the peri-lesion area of the striatum in ischemic mice revealed that microglia/macrophages (Fig. 5e, f) or astrocyte (Fig. 5g, h)

specific MEGF10$^{KO}$ or MERTK$^{KO}$ mice had elevated levels of SYP and Homer-1. Similarly, 14 days after hemorrhagic stroke, microglia/macrophage-specific MEGF10$^{KO}$ or MERTK$^{KO}$ mice also exhibited fewer Homer-1$^+$ and SYP$^+$ synaptic puncta in microglia/macrophages than control mice (Fig. 5l, m). Conditional MEGF10$^{KO}$ or MERTK$^{KO}$ in astrocytes did not affect the number of synaptic puncta engulfed by astrocytes (Fig. 5n, o). This phenomenon was also detected by western blot, as

**Fig. 2 Microglia/macrophages and astrocytes engulfed both excitatory and inhibitory synapses in stroke mice.** Colocalization assays showed microglia/macrophages (**a**, Iba-1[+], white) and astrocytes (**d**, GFAP[+], white) engulfed excitatory postsynaptic and inhibitory postsynaptic elements in the gliosis region at 14 days after MCAO. Arrows indicate mCherry-alone puncta within microglia/macrophages or astrocytes. Quantification comparing the volume of mCherry-alone synapses inside microglia/macrophages (**b**) or astrocytes (**e**). **b**, **e** Statistics are derived from 11 slices, $n = 4$ mice per group. Quantification showing the percentage of synaptic puncta inside/outside microglia/macrophages (**c**) or astrocytes (**f**) in the ischemic brain. Colocalization images showed mCherry-alone puncta within microglia/macrophages (**g**, white) or astrocytes (**j**, white) in the hemorrhagic brain. Quantification comparing the volume of mCherry-alone synapses inside microglia/macrophages (**h**) or astrocytes (**k**). Statistics are derived from 15, 12 slices (**h**) and 10, 10 slices (**k**) (from left to right), $n = 4$ mice per group. Quantification showing the percentage of synaptic puncta inside/outside microglia/macrophages (**i**) or astrocytes (**l**) in the hemorrhagic brain. Scale bar = 10 μm. *ExPost* excitatory postsynapse, *InhiPost* inhibitory postsynapse. *AU* arbitrary units. **b**, **e**, **h**, **k**, two-sided, unpaired Student's *t* test. Data are mean ± SD.

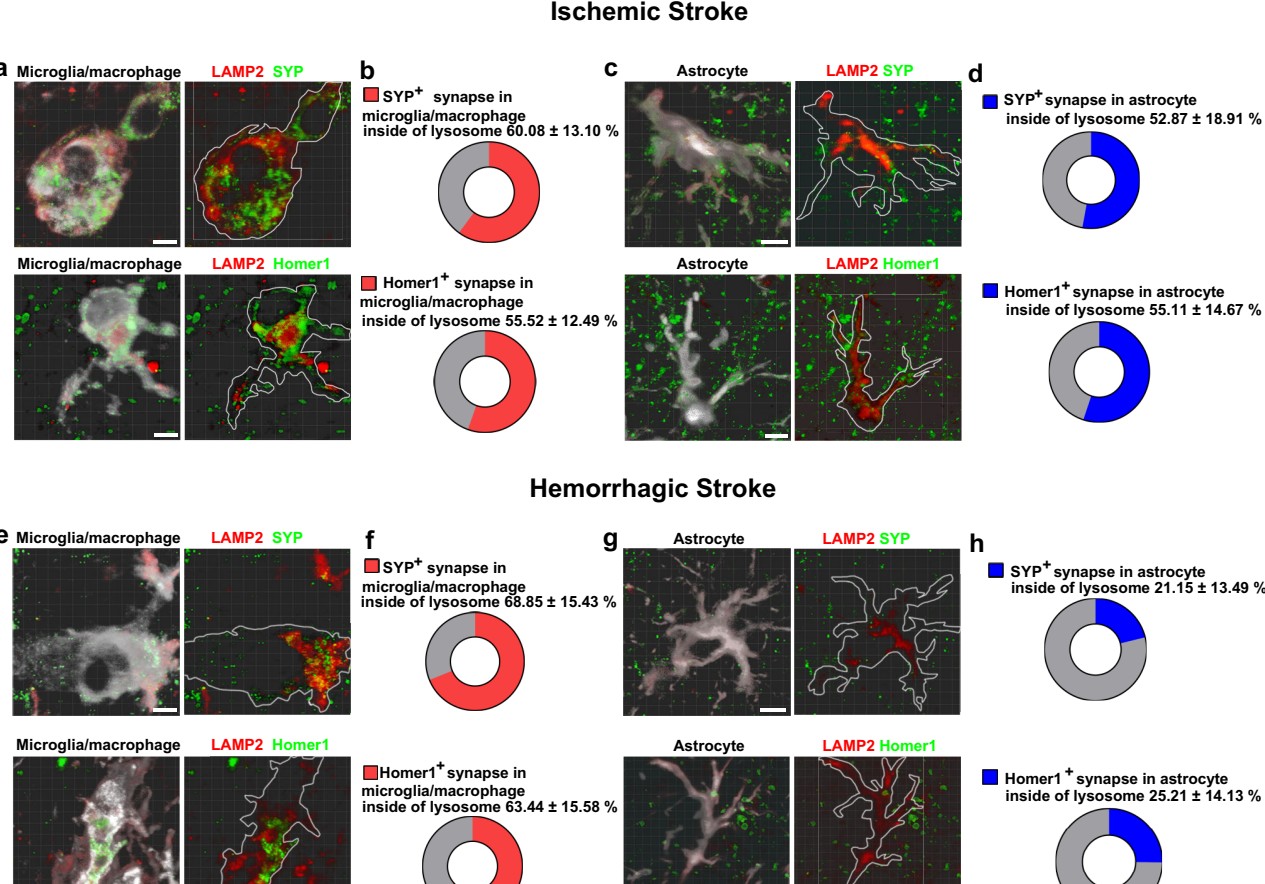

**Fig. 3 Microglia/macrophages and astrocytes digested synaptic elements through lysosome in stroke mouse brain. a–d** Representative images showed SYP[+] (green) and Homer-1[+] (green) signals in LAMP2[+] lysosomes (red) in Iba-1[+] microglia/macrophages (white) and GFAP[+] astrocytes (white) in the ischemic brain. **e–h** Representative images showed SYP[+] (green) and Homer-1[+] (green) signals in LAMP2[+] lysosomes (red) in Iba-1[+] microglia/macrophages (white) in the hemorrhagic brain, but rarely detected in GFAP[+] astrocytes (white). Bar = 10 μm. **a**, **c**, **e**, **g** $n = 4$ mice per group. Data are mean ± SD.

microglia/macrophage-specific MEGF10[KO] or MERTK[KO] mice had higher expression of SYP and Homer-1 (Fig. 5p, q), whereas astrocyte-specific MEGF10[KO] or MERTK[KO] mice had comparable expression of SYP and Homer-1 (Fig. 5r, s).

To evaluate the overall synapse density, we also colabeled pre- and postsynaptic markers and measured the density of SYP[+] presynapses, PSD95[+] postsynapses, and SYP[+]/PSD95[+] synapses, to show the overall synapse density. The densities of SYP[+] presynapse, PSD[+] postsynapse and SYP[+]/PSD95[+] synapse were increased in microglial and astroglial cKO mice after ischemic stroke, as compared with the control (Fig. 5i–k). In hemorrhagic mice, synapse density was increased in microglial cKO mice but not in astroglial cKO mice (Fig. 5t–v).

TEM combined with pre-embedding immunohistochemistry confirmed microglia/macrophage- and astrocyte-mediated phagocytosis of synapses. As shown in Fig. 6, after 14 days of ischemic stroke, both immunolabeled GFAP[+] reactive astrocytes and Iba-1[+] microglia/macrophages exhibited phagocytic inclusions, including synapses and myelin-like structures within their cytoplasm, indicating that microglia/macrophages and astrocytes actively engulf synapses. Furthermore, microglia/macrophage- or astrocyte-specific MEGF10[KO] or MERTK[KO] mice showed reduced engulfment ability compared to the control groups (Fig. 6a–d). At 14 days after hemorrhagic stroke, only Iba-1[+] microglia/macrophages exhibited phagocytic inclusions (Fig. 6e), while few synapses were detected in GFAP[+] astrocytes (Fig. 6g). Specific MEGF10[KO] or MERTK[KO] in

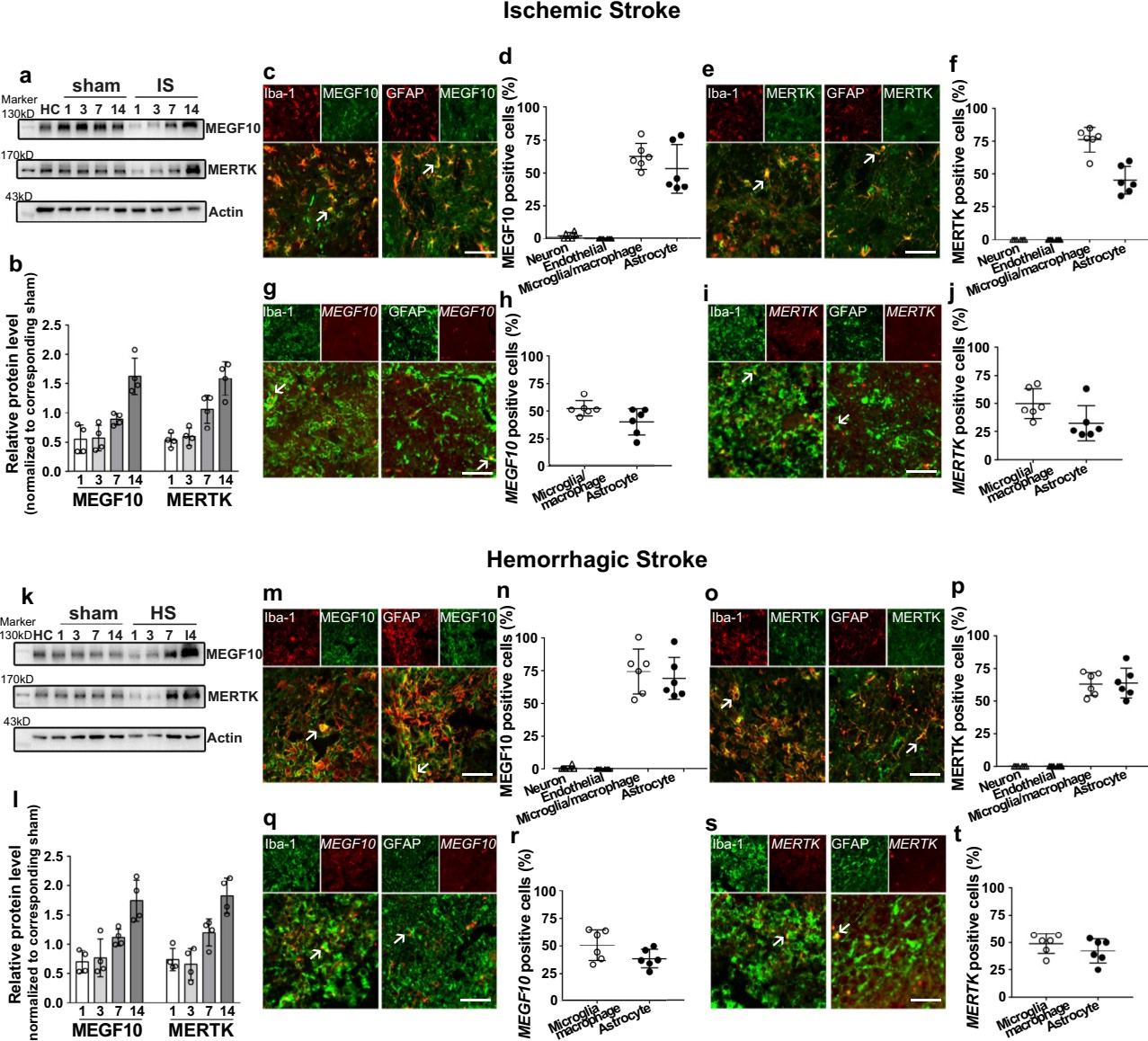

**Fig. 4 Upregulation of MEGF10 and MERTK levels in mouse brain following ischemic and hemorrhagic stroke.** Western blotting and quantification of MEGF10 and MERTK protein levels (normalized to corresponding sham) in ischemic (**a**) and hemorrhagic mouse brain (**k**). Immunostaining showed MEGF10 (green) and MERTK (green) were expressed in Iba-1+ microglia/macrophages (red) and GFAP+ astrocytes (red) in ischemic (**c–f**) and hemorrhagic stroke (**m–p**). Arrows indicated colocalization of MEGF10 and MERTK with microglia/macrophages and astrocytes. Bar = 50 μm. Fluorescence in situ hybridization study showed MEGF10 (red) and MERTK (red) signals colocalized in Iba-1+ microglia/macrophages (green) and GFAP+ astrocytes (green) in ischemic (**g–j**) and hemorrhagic stroke (**q–t**), indicated by arrows. Bar = 50 μm. (**b, l**) N = 4 mice per group. (**d, f, h, j, n, p, r, t**) Statistics are derived from six slices, n = 3 mice per group. Data are mean ± SD.

microglia/macrophages reduced synapse engulfment (Fig. 6f), whereas MEGF10^KO or MERTK^KO in astrocytes did not affect synapse engulfment (Fig. 6h).

**Conditional MEGF10 or MERTK knockout in microglia/macrophages or astrocytes increased spine density in the gliosis region.** To investigate the effects of MEGF10 and MERTK on dendritic spine structure after ischemic stroke, we performed Golgi-Cox staining to visualize neuronal dendritic spines in MEGF10^WT, MERTK^WT, C-MEGF10^KO, C-MERTK^KO, A-MEGF10^KO, and A-MERTK^KO mice. Based on the length and width of dendritic spines, they can be categorized as (1) Filopodia spine, with length >2 μm; (2) long-thin spine, with length <2 μm; (3) thin spine, with length <1 μm; (4) stubby spine, with the ratio of length/width <1; (5)

mushroom spine, with width >0.6 μm; and (6) branched spine, with two or more heads (Fig. 7a)[25]. The percentage of each dendritic spine type and the number of total spines, mature spines (mushroom, stubby, and branched), and filopodia spines (unstable, new-born spines) on secondary dendrites were calculated (Fig. 7b).

We found that in ischemic stroke, C-MEGF10^KO and C-MERTK^KO mice showed a significant increase in the number of total spines (2.2-fold and 3.0-fold, respectively), mature spines (1.9-fold and 2.5-fold, respectively) and filopodia spines (3.5-fold and 2.9-fold, respectively), as compared with the control (Fig. 7c–g). Similar results were also detected in A-MEGF10^KO and A-MERTK^KO mice (Fig. 7c–g), suggesting that inhibition of microglia/macrophage- and astrocyte-mediated synaptic engulfment is beneficial for promoting dendritic spine structure after ischemic stroke. In hemorrhagic stroke,

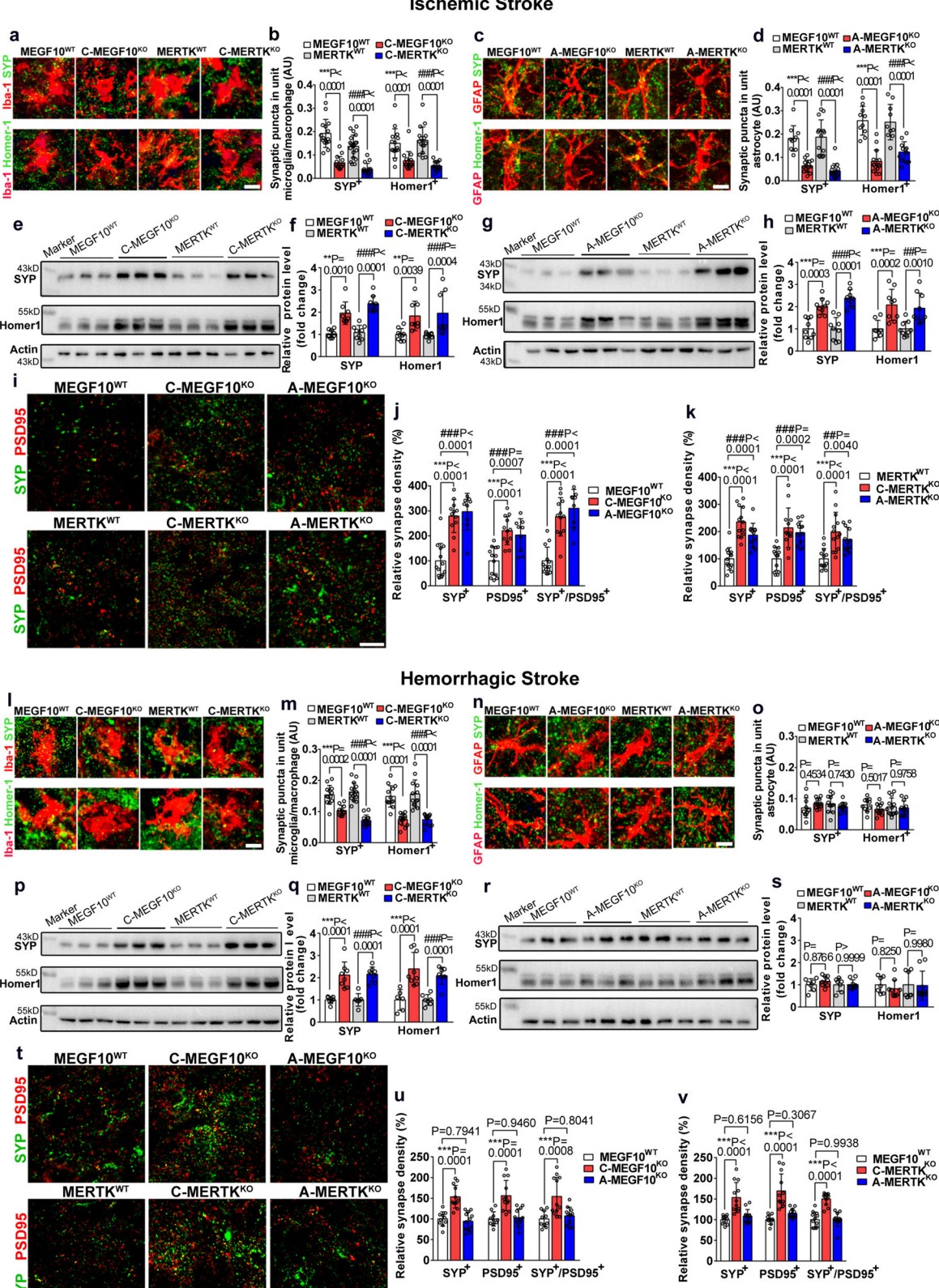

specific MEGF10[KO] and MERTK[KO] in microglia/macrophages increased the number of total dendritic spines and mature spines in the gliosis region (≈1.5-fold change, Fig. 7h–l). However, specific MEGF10[KO] and MERTK[KO] in astrocytes did not affect the number of spines (Fig. 7h, l), suggesting that microglia/macrophages but not astrocytes played a critical role in engulfing dendritic spines via MEGF10 and MERTK after hemorrhagic stroke.

**Conditional MEGF10 or MERTK knockout in microglia/macrophages or astrocytes differentially affected brain impairment and**

**Fig. 5 Conditional MEGF10 or MERTK knockout increased both pre- and postsynaptic proteins in stroke mouse brain. a–d** Representative images and quantification of the number of SYP$^+$ (green) and Homer-1$^+$ synapses (green) that engulfed by microglia/macrophages (red) or astrocytes (red) of MEGF10$^{WT}$, MERTK$^{WT}$, C-MEGF10$^{KO}$, C-MERTK$^{KO}$, A-MEGF10$^{KO}$, and A-MERTK$^{KO}$ mice at 14 days of ischemic stroke, respectively. Bar = 10 μm. Statistics are from 15, 15, 20, 15, 15, 15, 17, 15 slices (**b**) and 10, 15, 14, 15, 10, 15, 10, 12 slices (**d**) (from left to right), $n = 4$ mice per group. **e–h** Western blotting and quantification of SYP and Homer-1 levels in control and knockout mice following MCAO. $N = 9$, 8, 9, 9, 9, 8, 9, 8 mice (**f**) and 9, 9, 9, 9, 8, 9, 9, 9 (**h**) mice (from left to right). **i–k** Representative images and quantification of the synapse density of SYP$^+$ (green) and PSD95$^+$ synapses (red) in ischemic brain. Bar = 10 μm. Statistics are from 13, 12, 9, 13, 12, 9, 13, 12, 9 slices (**j**) and 13, 13, 12, 13, 13, 12, 13, 13, 12 slices (**k**) (from left to right), $n = 4$ mice per group. **l–o** Representative images and quantification of the number of SYP$^+$ (green) and Homer-1$^+$ synapses (green) that engulfed by microglia/macrophages (red) or astrocytes (red) in MEGF10$^{WT}$, MERTK$^{WT}$, C-MEGF10$^{KO}$, C-MERTK$^{KO}$, A-MEGF10$^{KO}$ and A-MERTK$^{KO}$ mice at 14 days of hemorrhagic stroke, respectively. Bar = 10 μm. Statistics are derived from 13, 11, 15, 13, 13, 13, 13, 13 (**m**) and 10, 12, 12, 13, 10, 12, 13, 12 (**o**) (from left to right), $n = 4$ mice per group. **p–s** Western blotting and quantification of SYP and Homer-1 levels in control and knockout mice following hemorrhagic stroke. $N = 7$, 8, 7, 8, 6, 9, 7, 8 mice (**q**) and 7, 9, 7, 9, 7, 9, 7, 9 mice (**s**) (from left to right). **t–v** Representative images and quantification of the synapse density of SYP$^+$ (green) and PSD95$^+$ synapses (red) in hemorrhagic brain. Bar = 10 μm. Statistics are derived from 10, 12, 12, 10, 12, 12, 10, 12, 12 slices (**u**) and 11, 12, 12, 10, 12, 12, 12, 12, 12 slices (**v**) (from left to right), $n = 4$ mice per group. For all quantification data, two-way ANOVA followed by Tukey's test. Data are mean ± SD.

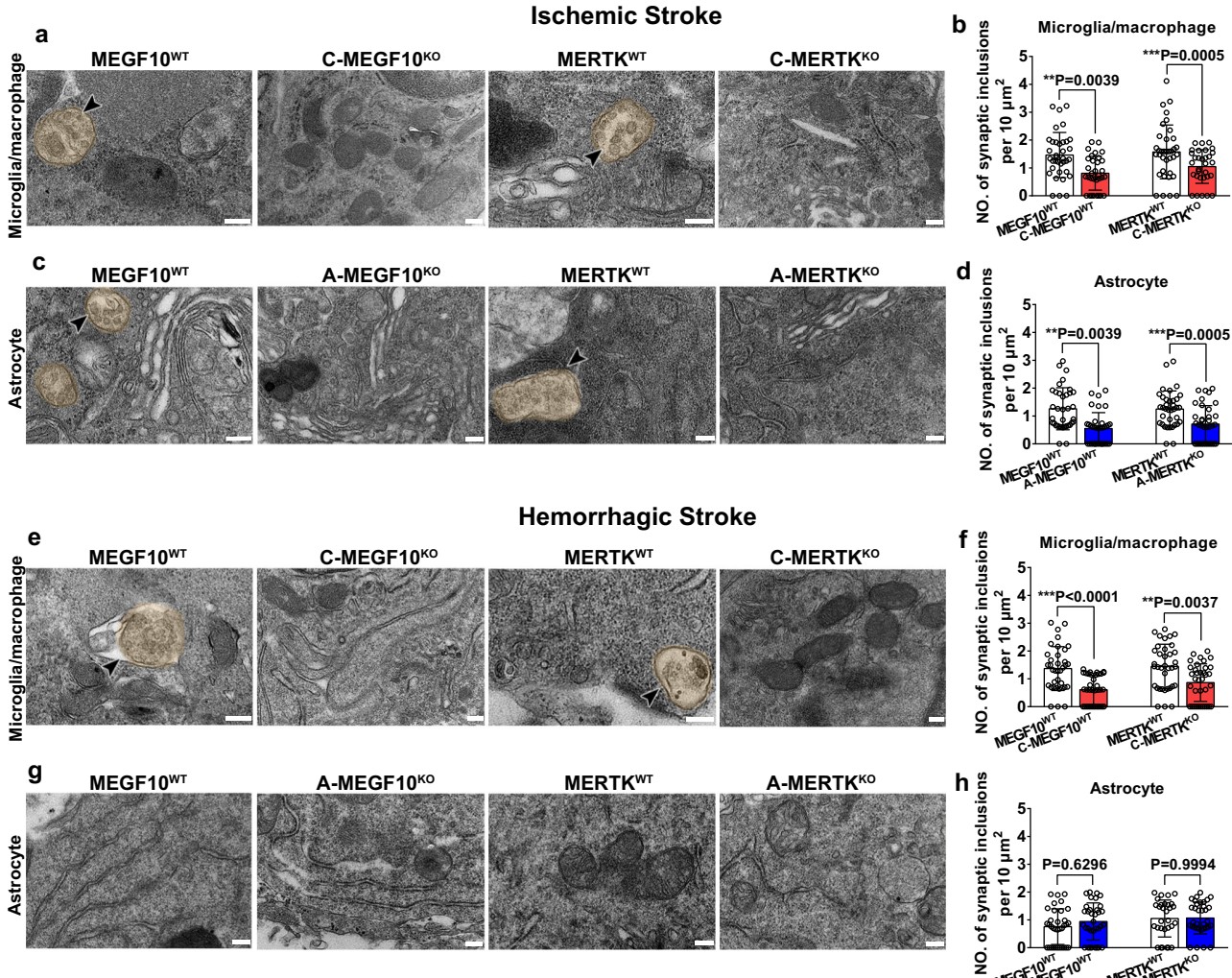

**Fig. 6 Conditional MEGF10 or MERTK knockout reduced glial cell-mediated synapse engulfment after stroke. a–d** TEM images showed Iba-1$^+$ microglia/macrophages and GFAP$^+$ astrocytes in the gliosis region enwrapped engulfed synaptic elements in MEGF10$^{WT}$ and MERTK$^{WT}$ mice, but the number of engulfed synaptic elements were reduced in C-MEGF10$^{KO}$, C-MERTK$^{KO}$, A-MEGF10$^{KO}$, and A-MERTK$^{KO}$ mice after ischemic stroke. Arrowheads indicate synaptic elements that were engulfed by microglia/macrophages (**a**, **b**) or astrocytes (**c**, **d**). Statistics are derived from 36, 32, 36, 31 cells (**b**) and 36, 36, 36, 36 cells (**d**) (from left to right), $n = 3$ mice per group. **e–h** TEM images showed Iba-1$^+$ microglia/macrophages in the gliosis region contained synaptic elements in MEGF10$^{WT}$ and MERTK$^{WT}$ mice (**e**, **f**); while synaptic elements were rarely detected in GFAP$^+$ astrocytes (**g**, **h**). Statistics are derived from 36, 36, 36, 36 cells (**f**) and 34, 34, 30, 34 cells (**h**) (from left to right), $n = 3$ mice per group. Bar = 200 nm. One-way ANOVA followed by Tukey's test. Data are mean ± SD.

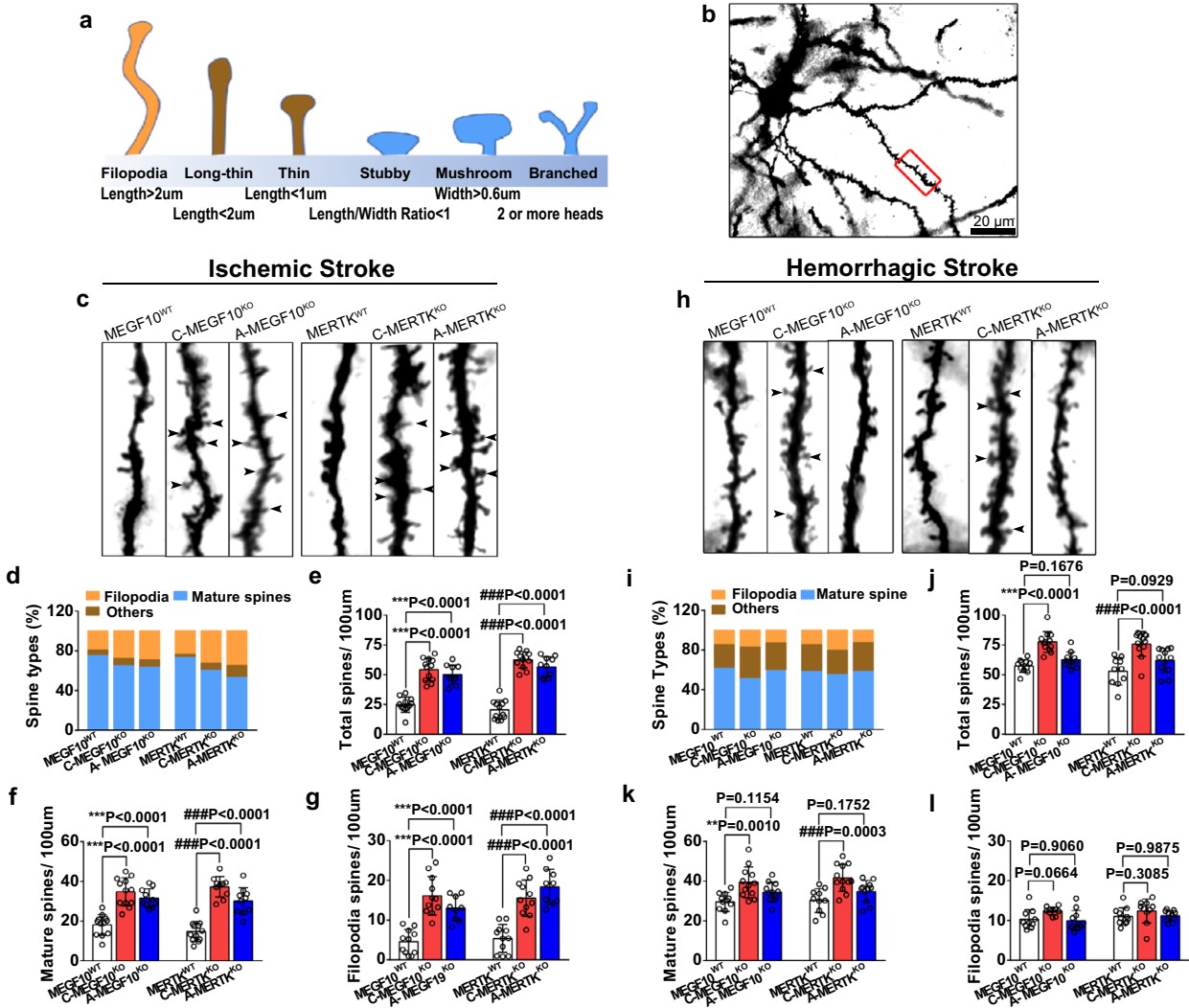

**Fig. 7 Conditional knockout of MEGF10 and MERTK increased dendritic spine density in stroke mice. a** Schematic diagram of different morphology of dendritic spines. **b** Low magnification of Golgi staining images of neuron in the gliosis region. The experiment was repeated at least three times independently. Scale bar = 20 μm. The red box indicates the region of high magnification of spines in the secondary dendrite as shown in **c** and **h**. Representative images of dendritic spines and bar graph showed the percentage of each spine types (**d, i**) and the number of total spines (**e, j**), mature spines (including stubby, mushroom, and branched spines) (**f, k**) and filopodia spines (**g, l**) of MEGF10^WT, MERTK^WT, C-MEGF10^KO, C-MERTK^KO, A-MEGF10^KO, A-MERTK^KO mice at 14 days after ischemic (**d–g**) and hemorrhagic stroke (**i–l**). Statistics are derived from 12, 12, 10, 12, 12, 10 slices (**e**), 12, 12, 10, 12, 10, 10 slices (**f**) and 11, 11, 9, 11, 11, 9 slices (**g**) (from left to right), n = 4 mice per group. Statistics are derived from 11, 12, 11, 10, 12, 12 slices (**j**), 11, 12, 11, 10, 12, 11 slices (**k**), and 10, 10, 11, 11, 10, 10 slices (**l**) (from left to right), n = 4 mice per group. One-way ANOVA followed by Dunnett's test. Data are mean ± SD.

**behavioral outcomes in mice after ischemic and hemorrhagic stroke.** To explore the effects of microglia/macrophage- and astrocyte-mediated synapse engulfment on the outcomes of mice after ischemic stroke, we examined the brain atrophy volume and neurobehavioral function in MEGF10^WT, MERTK^WT, C-MEGF10^KO, C-MERTK^KO, A-MEGF10^KO, and A-MERTK^KO mice after MCAO. Cresyl violet staining results showed that conditional MEGF10^KO in microglia/macrophages or astrocytes reduced the brain atrophy volume 14 days after MCAO (Fig. 8a, b), and specific MERTK^KO in microglia/macrophages and astrocytes also reduced the brain atrophy volume (Fig. 8c, d). Ventricular enlargement is a common symptom of hemorrhagic stroke[26,27]. To investigate the effects of microglia/macrophage- and astrocyte-mediated synapse engulfment on the outcomes of mice after hemorrhagic stroke, we measured the ventricular volume of MEGF10^WT, MERTK^WT, C-MEGF10^KO, C-MERTK^KO, A-MEGF10^KO and A-MERTK^KO mice at 14 days after hemorrhagic stroke. Conditional

MEGF10^KO or MERTK^KO in microglia/macrophages significantly ameliorated ventricular enlargement, while MEGF10^KO or MERTK^KO in astrocytes did not affect the ventricular volume (Fig. 8e–h).

Furthermore, microglia/macrophage- and astrocyte-specific MEGF10^KO and MERTK^KO mice performed better in the mNSS (Fig. 9a, b), rotarod test (Fig. 9c, d), and step-through passive avoidance test (Fig. 9e, f, g). Moreover, specific MEGF10^KO or MERTK^KO in microglia/macrophages improved neurobehavioral outcomes of mice after hemorrhagic stroke, while MEGF10^KO or MERTK^KO in astrocytes did not affect neurobehavioral outcomes (Fig. 9h–m).

**Reactive astrogliosis exhibited different expressions of phagocytosis-related genes between ischemic and hemorrhagic stroke.** To explain why astrocytes displayed different phagocytic features between ischemic and hemorrhagic stroke, the transcriptional profile differences between ischemic and hemorrhagic

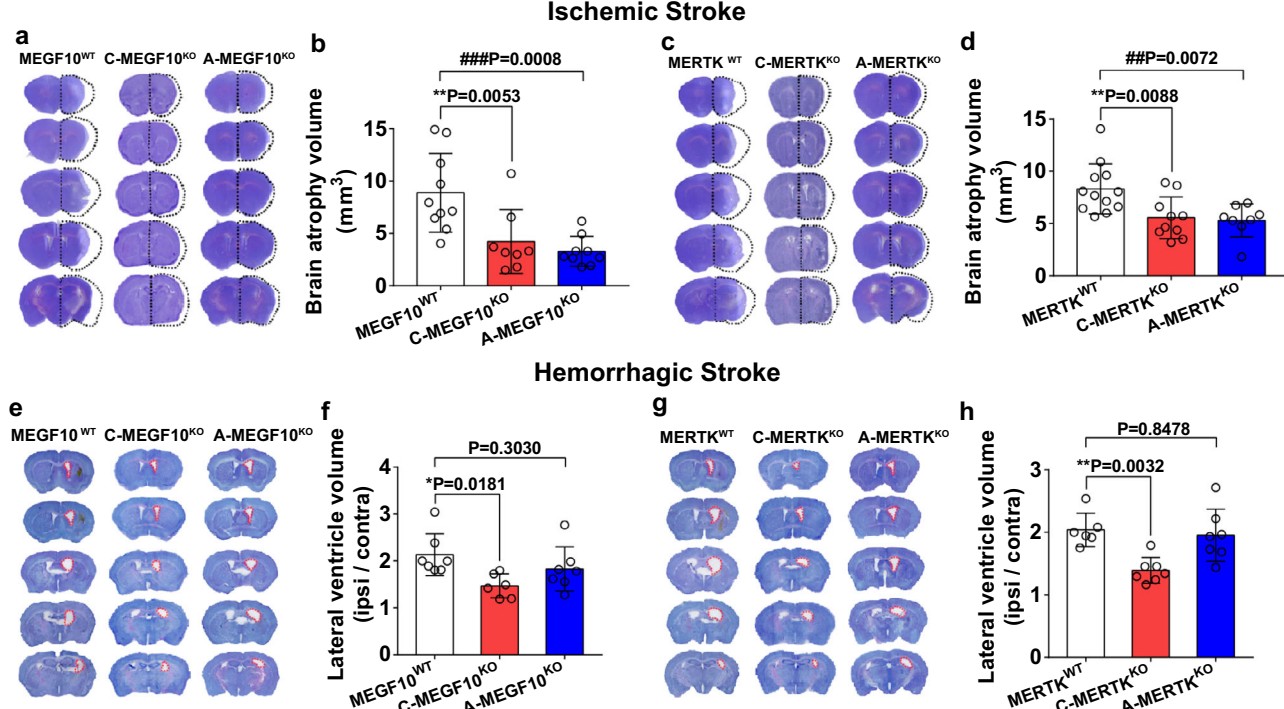

**Fig. 8 Conditional MEGF10 or MERTK knockout in microglia/macrophages or astrocytes differentially affected brain impairment in mice after ischemic and hemorrhagic stroke. a–d** Cresyl violet-stained brain sections and quantification of atrophy volume of MEGF10[WT], C-MEGF10[KO], A-MEGF10[KO], MERTK[WT], C-MERTK[KO], and A-MERTK[KO] mice at 14 days after ischemic stroke. Black dashed lines indicate brain atrophy. $N = 10, 8, 9$ mice (**b**) and 12, 10, 8 mice (**d**) (from left to right). **e–h** Cresyl violet-stained brain sections and quantification of lateral ventricle volume of hemorrhagic mice. Red dashed lines indicate lateral ventricle of ipsilateral brain. $N = 7, 6, 7$ mice (**f**) and 6, 7, 7 mice (**h**) (from left to right). One-way ANOVA followed by Dunnett's test. Data are mean ± SD.

brains were examined using single-cell RNA sequencing (scRNA-seq). Cluster analysis using nonlinear dimensionality reduction (*t*-distributed stochastic neighbor embedding [tSNE]) revealed the differences in global gene expression profiles of healthy and injured striatum and identified clusters of cells with unique genetic signatures in both ischemic brain and hemorrhagic brain (Fig. 10a). Genes with a *p* value <0.05 and fold change ≥1.5 were regarded as differentially expressed genes (DEGs). For astrocytes, 135 DEGs were downregulated in hemorrhagic stroke compared to ischemic stroke, and we further conducted Gene Ontology (GO), and Kyoto Encyclopedia of Genes and Genomes (KEGG) pathway analyses. The top fold-change genes are shown in the heatmap, and we found that some phagocytosis-related genes such as *Hspa1a*, *Vim*, and *Mt1*, were downregulated in astrocytes in the hemorrhagic brain compared to the ischemic brain (Fig. 10b). MEGF10 and MERTK were also detected in astrocytes of mice after ischemic stroke and hemorrhagic stroke (Fig. 10c).

In addition, phagocytosis-related biological processes, as well as cellular components including phagocytic vesicles and synaptic structures were significantly downregulated in astrocytes in the hemorrhagic brain (Fig. 10d). The secondary profiling of astrocytic subtypes yielded 10 different subtypes with distinct functional cell identities (Fig. 10e). Specific top gene markers representing each subcluster were found across major astroglial functions, including synapse function/plasticity (*C1ql2*[28,29], *Agt*[30]), neurotransmission (*Slc39a12*[31,32]), gap junction (*Gjb6*[33,34]), and ion modulation/binding (*Kcnj8*[35], *Pln*[36]) (Fig. 10f). Among all the subtypes, the proportion of subtype 3 astrocytes in ischemic mice was ~20%, while in hemorrhagic mice, the proportion was <2% (Fig. 10e). This cluster may be responsible for the different phagocytic features of astrocytes between ischemic stroke and hemorrhagic stroke. Thus, we analyzed 881 marker genes (*p* value <0.01) of subtype 3 using

Metascape (www.metascape.org). As expected, subtype 3 exhibited a notable upregulation of synapse pruning related processes, aligned with abundant synaptic structures and lysosomes in cells (Fig. 10g).

For microglia, the tSNE map indicated that the distribution and proportion of microglia/macrophages were very similar in the ischemic and hemorrhagic stroke models (Supp. Fig. 7a). We obtained 75 DEGs in total (hemorrhagic stroke vs. ischemic stroke, 54 were upregulated, 21 were downregulated) and performed an analysis in the same way. The top fold-change genes are shown in the heatmap (Supp. Fig. 7b). MEGF10 and MERTK were also detected in microglia of mice after ischemic stroke and hemorrhagic stroke (Supp. Fig. 7c). The bar graph showed that these DEGs entail many immunity-related biological changes, but a few phagocytosis-related processes (Supp. Fig. 7d, e), supporting the finding that microglia in ischemic or hemorrhagic brains share similar phagocytic patterns.

## Discussion

In this study, we revealed that at the poststroke repair and remodeling stage, reactive microgliosis and astrogliosis were active in engulfing synapses through MEGF10- and MERTK-related pathways, but showed different temporal phagocytic features in ischemic and hemorrhagic stroke. Our loss-of-function experiments further demonstrated that reactive microgliosis- and astrogliosis-mediated synapse engulfment differentially influenced the neurobehavioral outcomes of ischemic and hemorrhagic stroke mice. We found that inhibition of microgliosis- or astrogliosis-mediated synapse engulfment could improve the outcomes of ischemic stroke mice; while in hemorrhagic stroke, suppression of microgliosis- but not astrogliosis-mediated

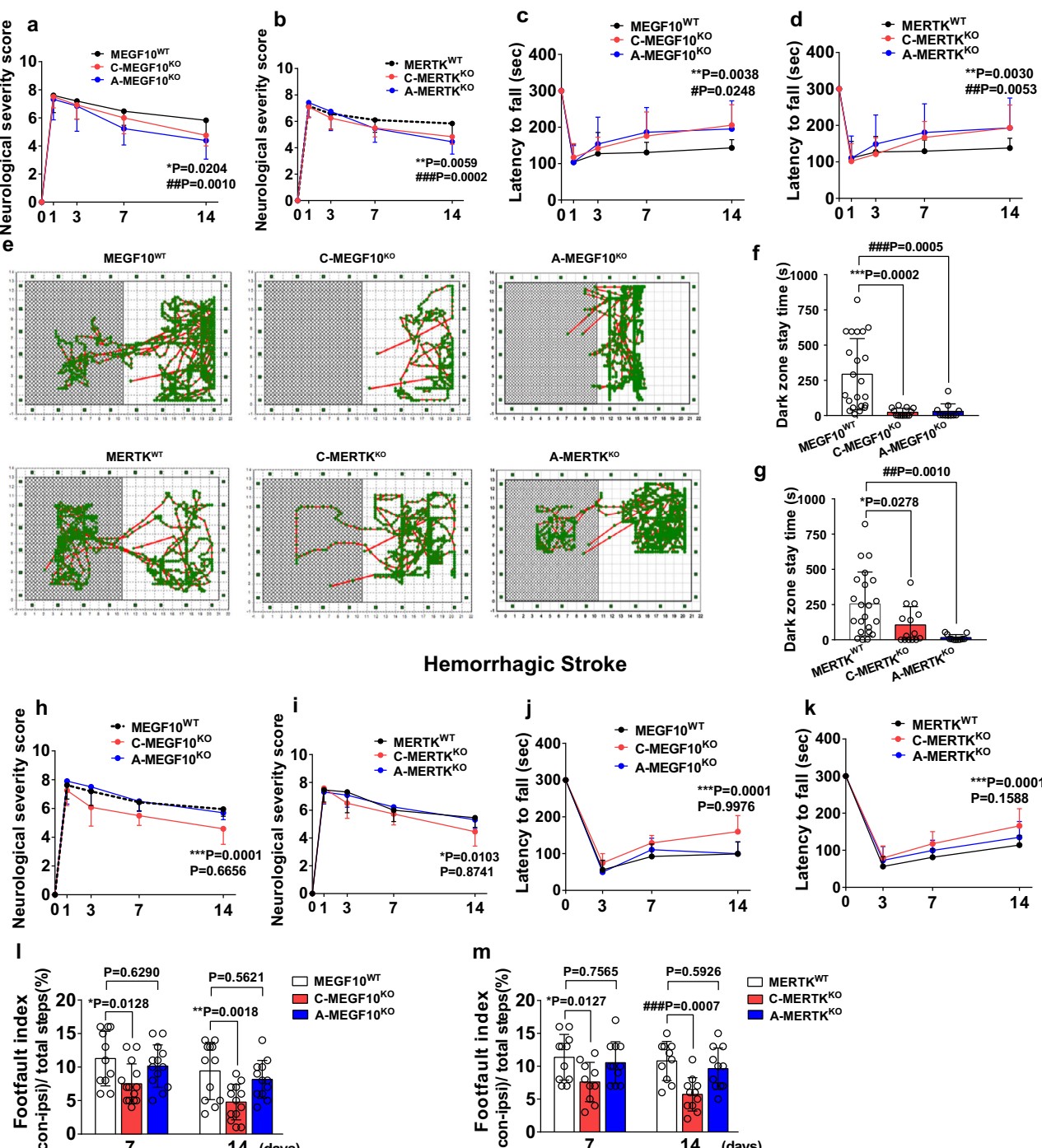

**Fig. 9 Conditional MEGF10 or MERTK knockout in microglia/macrophages or astrocytes differentially affected behavioral outcomes in mice after ischemic and hemorrhagic stroke.** mNSS (**a, b**) and rotarod test (**c–d**) were performed to examine the neurobehavioral outcomes of ischemic mice. *, MEGF10$^{WT}$ VS C-MEGF10$^{KO}$, MERTK$^{WT}$ VS C-MERTK$^{KO}$; #, MEGF10$^{WT}$ VS A-MEGF10$^{KO}$, MERTK$^{WT}$ VS A-MERTK$^{KO}$. $N = 17, 12, 13$ mice (**a**); 19, 12, 11 mice (**b**); 11, 12, 10 mice (**c**); and 19, 13, 11 mice (**d**) (day 14, from left to right). **e** Representative images showed travel patterns in the smart cage. **f, g** Quantification of the time that mice of different groups stayed in the dark zone. $N = 22, 17, 17$ (**f**) and 22, 16, 15 mice (**g**) (from left to right). mNSS (**h–i**), rotarod test (**j–k**), and grid-walking test (**l–m**) were used to examine neurobehavioral outcomes of hemorrhagic mice at different time points. **h–k** *, MEGF10$^{WT}$ VS C-MEGF10$^{KO}$, MERTK$^{WT}$ VS C-MERTK$^{KO}$; &, MEGF10$^{WT}$ VS A-MEGF10$^{KO}$, MERTK$^{WT}$ VS A-MERTK$^{KO}$. $N = 21, 12, 10$ mice (**h**); 11, 11, 13 mice (**i**); 12, 14, 11 mice (**j**); and 11, 14, 11 mice (**k**) (day 14, from left to right). $N = 11, 14, 13, 11, 13, 12$ mice (**l**) and 11, 10, 11, 10, 10, 11 mice (**m**) (from left to right). **a–d**, **h–m** Two-way ANOVA followed by Tukey's test; **f, g** one-way ANOVA followed by Tukey's test. Data are mean ± SD.

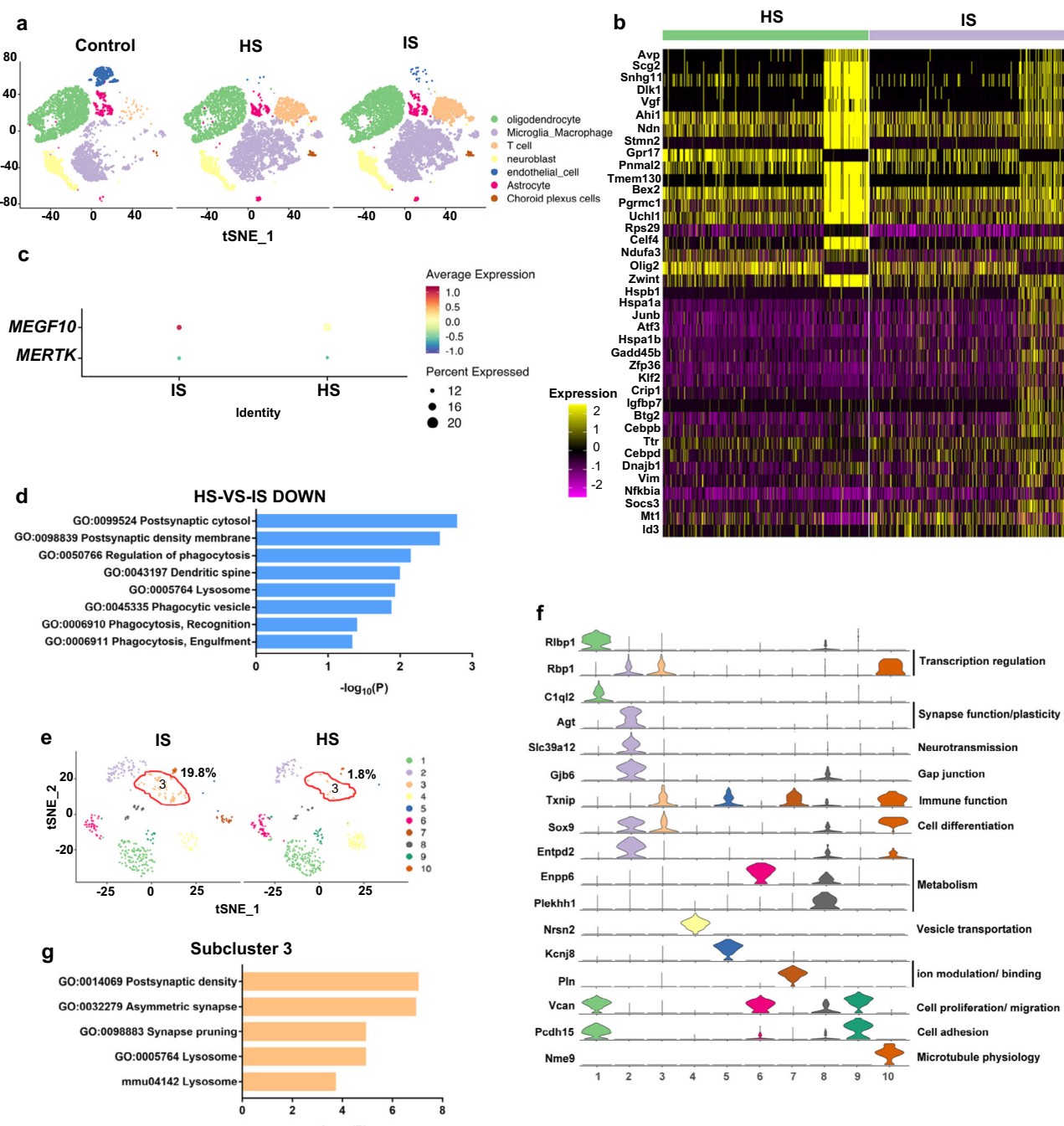

**Fig. 10 scRNA-seq revealed phagocytosis-related gene expression difference of astrocytes between ischemic and hemorrhagic stroke. a** tSNE map showed the expression profiles of the striatum in control, ischemic and hemorrhagic mice, respectively. **b** Heatmap showed fold change of top 20 genes. **c** Dot plot showing MEGF10 and MERTK expression in astrocytes. **d** Bar chart showed phagocytosis-related GO and KEGG pathways that downregulated in hemorrhagic stroke (HS), as compared with ischemic stroke (IS). Representative terms were shown in rows and $-\log_{10}$ (p) in columns. **e** tSNE map showed subclusters of astrocytes in IS and HS. **f** Violin plots showed top marker genes in specific astrocyte subclusters. **g** Bar chart showed phagocytosis-related GO and KEGG terms enriched in subcluster 3 of astrocytes.

phagocytosis was beneficial for neurobehavioral outcomes. Furthermore, by comparing the transcriptomics of ischemic and hemorrhagic mouse brains, we found that phagocytosis-related genes and biological processes were downregulated in astrocytes after a hemorrhagic stroke.

At the early stage of stroke, microglia are actively involved in engulfing degenerating neuron debris[37,38]. Recently, phagocytic astrocytes were also observed within the ischemic perifocal region and engulfed dead neurons at 7 days after ischemic stroke,

whereas phagocytic microglia were mainly observed in the ischemic core region at 3 days after ischemia[16]. In our study, we investigated the phagocytosis of microglia/macrophages and astrocytes at both the acute and subacute stages of ischemic and hemorrhagic stroke. We found that Mac-2 was upregulated as early as 1 day following stroke, but without changes of MEGF10 and MERTK protein levels. We also found that synapses were engulfed by microglia/macrophages and astrocytes at 1 day after stroke (Supp. Fig. 8), suggesting that MEGF10 and MERTK did

not play critical roles in mediating early synapse removal following stroke. Complement component-dependent mechanisms may be responsible for early synapse engulfment following stroke, which is supported by a previous study. For example, Alawieh et al.[39] showed that the complement system is activated by ischemic stroke, and specific local complement inhibition reduced cell death and inflammation induced by stroke.

Normal astrocytes are enriched in MEGF10 and MERTK, and normal microglia are enriched in MERTK[40]. Our immunostaining and in situ hybridization results demonstrated that MEGF10 and MERTK were upregulated in both reactive microgliosis and astrogliosis after ischemic or hemorrhagic stroke. This observation is surprising since a previous study showed that MEGF10 was not expressed in Iba-1[+] immune cells after ischemic stroke[16]. One potential explanation for this difference is that, unlike the 15-min stroke model they used, in the present study, we used a 90-min stroke model, which caused more severe injury, and activation of glia and immune cells. In addition, we also observed that MEGF10 and MERTK were expressed in microglia/macrophages and astrocytes in the hemorrhagic stroke human brain (Supp. Fig. 9). MEGF10 and MERTK protein level was notably increased at 14 day following stroke. Interestingly, in our scRNA-seq data, we did not observe a significant difference in the expression of MEGF10 and MERTK in astrocytes (Fig. 10c) or microglia/macrophages (Supp. Fig. 7c) among the three groups, which is inconsistent with our western blot results. The possible reason is that single-cell RNA sequencing is a genomic approach for the detection and quantitative analysis of messenger RNA molecules in each cell, while western blotting is a method used to detect specific proteins from a mixture of proteins. The expression of mRNA and proteins is time-dependent, and they may have different expression profiles at the same time. Genetic MEGF10[KO] or MERTK[KO] in microglia/macrophages or astrocytes increased the levels of pre- and postsynaptic proteins at 14 days after ischemic stroke. However, in hemorrhagic stroke, a similar phenomenon was not detected in astrocyte-specific knockout mice. These differences may be attributed to the low number of phagocytic astrocytes in the gliosis region of hemorrhagic stroke, even though MEGF10 and MERTK were upregulated in astrocytes.

Gliosis and glial scar formation are fundamental pathophysiologies of ischemic and hemorrhagic stroke that critically affect brain remodeling and neural regeneration[41]. Extensive evidence supports the view that gliosis inhibits axon extension and brain functional recovery by secreting inhibitory factors such as chondroitin sulfate proteoglycans and bone morphogenetic proteins[42]. In contrast, studies performed by the Sofroniew laboratory over the past decade have demonstrated that preventing gliosis formation following CNS injury does not result in increased regeneration[6]. In our study, synapse engulfment was detected in reactive microgliosis and astrogliosis, leading us to speculate that inhibition of this process would reduce synapse loss and promote synapse reconnection in stroke mice. Indeed, inhibiting synapse engulfment by glial cells increased the levels of both pre-and postsynapses, as well as the number of dendritic spines, and improved neurobehavioral outcomes in ischemic stroke mice. Similar results were also detected when specifically inhibiting microglia/macrophage- but not astrocyte-mediated synapse engulfment in hemorrhagic mice.

Such differences could be attributed to two reasons. First, astrocytes respond differentially to different pathological microenvironments in ischemic and hemorrhagic stroke. A recent study showed that astrocytes alone showed mild reactiveness under hemorrhagic conditions in vitro[43], and astrocytes secreted cytokines to cross-talk with microglia following intracerebral hemorrhage[44]. These findings suggested that astrocytes in the hemorrhagic brain played an indirect role in regulating neuropathology. Herein, we demonstrated that ~50% of astrocytes are phagocytic and engulf a large number of synapses in the gliosis region at 14 days after ischemic stroke, whereas only 5% of astrocytes are phagocytic in hemorrhagic stroke and few synapses are engulfed by astrocytes. Second, astrocytes displayed different profiles of phagocytosis-related genes between ischemic and hemorrhagic mice, as revealed by scRNA-seq. Specifically, phagocytosis-related genes such as *Hspa1a*, *Vim*, and *Mt1*, and phagocytosis-related biological processes, as well as cellular components, including phagocytic vesicles and synaptic structures, were downregulated in astrocytes in the hemorrhagic brain. Further insight analysis showed 10 different subtypes of astrocytes with distinct functional cell identities. Notably, subtype 3, which specifically expressed *Txnip* (immune-related)[45,46], *Rbp1* (transcription regulation-related)[47] and *Sox9* (astrocyte differentiation-related)[48–50], attracted our attention as the proportion of subtype 3 in hemorrhagic mice was much lower than that in ischemic mice. Subtype 3 astrocyte exhibited a notable upregulation of synapse pruning related processes, aligned with abundant synaptic structures and lysosomes in the cytostome, which may be responsible for the different phagocytic features between ischemic and hemorrhagic stroke. Through genetic analysis, we found that subtype 3 was abundant with C1qa and C1qb, two subunits of complement C1q[51]. Previous studies have shown that C1q localizes to synapses in the developing retina and brain, and plays critical roles important for synapse refinement[12]. In Alzheimer's disease, C1q is upregulated and deposited onto synapses[13,52]. Together, we assume that *Txnip*[+]/*Rbp1*[+]/*Sox9*[+] astrocytes could engulf C1q[+] synapses at the recovery stage of stroke. As the number of subtypes 3 was negligible in the hemorrhagic brain, this may account for why astrocyte-mediated synapse engulfment was rarely detected in hemorrhagic stroke. We also analyzed whether there were different inflammatory/immune cell compositions in ischemic and hemorrhagic stroke. Our data showed that the proportion and number of T cells in the hemorrhagic brain were 14.60% and 1250, respectively, and those in the ischemic brain were 14.10% and 1769, respectively. Considering that the composition of T cells in these two experimental groups was similar, we assume that inflammatory/immune cells may have a negligible impact on astrocytes.

Recovery from neural injury depends on several critical conditions, including neuroprotection after acute brain injury and local remodeling of synapses, axons, and circuits in the chronic stage. Our study demonstrated that knockout of MEGF10 or MERTK improved neurobehavioral recovery as a result of inhibiting synapse engulfment, which raises the question of whether these genetic manipulations also influence cell death and axonal regeneration. By analyzing TUNEL[+] apoptotic cells, we found that a lack of MEGF10 and MERTK did not affect cell death after stroke. By quantifying SMI32[+]/MBP[+]-damaged myelinated axons, we concluded that knockout of MEGF10 or MERTK decreased axonal degeneration in the ischemic mouse brain but not in the hemorrhagic mouse brain, suggesting that in addition to protecting synapses, genetic manipulation of MEGF10 or MERTK also reduces axonal degeneration in ischemic stroke (Supp. Fig. 10).

The heterogeneous activation of astrocytes should be taken into account to further develop effective treatments targeting glial cell-mediated synapse engulfment. Recent findings revealed that stroke-induced different types of astrocytes, including neurotoxic A1 astrocytes and neuroprotective A2 astrocytes[53,54]. Liddlelow et al.[54] demonstrated that A1 reactive astrocytes displayed deficient synapse engulfment in the developmental lateral geniculate nucleus. However, according to our scRNA-seq results, we found that astrocytes in the ischemic brain had more A1 properties,

whereas astrocytes in the hemorrhagic brain were neither A1-like nor A2-like (Supp. Fig. 11), suggesting that the differences in astrocyte engulfment between the two-stroke models are not A1 or A2 phenotype dependent. The different testing time points following CNS injury could account for this difference, since Liddlelow et al. examined synapse engulfment by astrocyte 24 hrs following acute cerebral ischemia, whereas we focused on the late stage of brain injury. More importantly, accumulating evidence argued that the using A1/A2 concept to identify specific astrocyte functional subtypes was untested and potentially misleading[55], since multiple studies have found that few and even none of the A1 marker genes are detectably expressed in various neurode-generative disease and brain injury[56,57]. In line with these studies, our results indicated the lack of A1/A2 signature genes in reactive astrogliosis following distinct stroke models. Further efforts are needed to gain a better understanding of astrocyte reactivity which may vary in context-dependent manners.

Our study demonstrated that inhibition of synapse engulfment was beneficial for improving neurobehavioral outcomes, para-doxically, this process may also prevent aberrant synapses from being eliminated. It is known that pruning unnecessary synapses by glial cells is critical for normal brain development[58]. Thus, one concern of our study is that inhibition of synapse engulfment may cause aberrant synapse formation after injury, which could con-tribute to the pathogenesis of stroke, such as seizures. We did not detect any abnormal behavior of mice that lacked MEGF10 or MERTK, at least within 14 days after stroke, suggesting the dif-ferent mechanisms of de novo synapse formation in the devel-oping brain and synapse remodeling after injury. Long-term observation is required to further determine whether inhibition of MEGF10 or MERTK would cause abnormal symptoms and behavior in mice. In addition, future efforts are required to determine whether inhibiting phagocytosis of reactive gliosis saves functional or impaired synapses.

## Methods

### Generation of loxp floxed MEGF10 and MERTK mice.
MEGF10-flox mice were generated as previously reported[23]. In brief, a targeting construct was generated by the trans-NIH Knock-Out Mouse Project (KOMP) and obtained from the KOMP Repository (www.komp.org). C57BL/6 embryonic stem cells (ESCs) were used for electroporating the MEGF10 conditional KO construct and the correctly targeted ESCs were screened by Southern blot. Resulted chimera males were bred with C57BL/6 wild-type females to obtain MEGF10-flox founder mice. All experiments were performed with support from the Stanford Transgenic Knockout and Tumor Model Center (TKTC).

MERTK-floxed mice were generated using clustered regularly interspaced short palindromic repeats technology (Applied StemCell) as noted in a separate work (Park et al., in preparation). In brief, a mixture of two sets of active guide RNA molecules, two single-stranded oligodeoxynucleotides, and appropriate Cas9 mRNA were injected into the cytoplasm of C57BL/6 embryos, and two LoxP cassettes were inserted into intron 1 and intron 2 to flank exon 2 of the MERTK locus. The following primers were used for genotyping:

forward: 5′-CTTCATCATGCTCACCTCAAACC-3′,
reverse: 5′-GTGCAGAATATTCACCTGACTGC-3′.

To specifically knockout MEGF10 or MERTK in microglia/macrophages or astrocytes, loxp floxed MEGF10, or loxp floxed MERTK mice were crossed with Aldh1l1 Cre-ERT2 or CX3CR1Cre-ERT2 mice (gift from Dr. Won-Suk Chung), and tamoxifen (Sigma-Aldrich, MO) dissolved in 95% corn oil at a final concentration of 10 mg/ml was intraperitoneally injected (0.075 mg/g body weight) once a day for 5 consecutive days. Mice were used at least 10 days after the final dose of tamoxifen.

### A mouse model of middle cerebral artery occlusion.
Animal studies were reported in accordance with Animal Research: Reporting in Vivo Experiments: ARRIVE guidelines. The procedure for using laboratory animals was approved by the Institutional Animal Care and Use Committee (IACUC) of Shanghai Jiao Tong University, Shanghai, China. The surgery of MCAO was performed as described previously[59]. Adult mice (10–12 weeks, $n = 30$ per group, male: female = 1:1) were anesthetized with 1.5–2% isoflurane and 30%/70% oxygen/nitrous oxide. Body temperature was maintained at 37 °C using a heating pad. In brief, the common carotid artery, internal carotid artery and external carotid artery were separated. A 6-0 suture (Covidien, Mansfield, MA) coated with silicon (Heraeus Kulzer,

Germany) was inserted from the external carotid artery, followed by the internal carotid artery, and gently stopped at the opening of middle cerebral artery. The success of occlusion was determined by monitoring the decrease in surface cerebral blood flow (CBF) to 10% of baseline CBF using a laser Doppler flowmetry (Moor Instruments, Devon, UK). Reperfusion was performed by withdrawing the suture 90 min after MCAO.

### Collagenase-induced intracerebral hemorrhage.
Adult mice (10–12 weeks, $n = 30$ per group, male: female = 1:1) were anesthetized with isoflurane and secured in a stereotactic frame (RWD Life Science co., Shenzhen, China), then subjected to intracerebral hemorrhage using collagenase IV (Sigma-Aldrich, MO) as previously described with modification[60]. A 29-gauge needle was inserted ste-reotaxically into the right striatum (coordinates: 0.5 mm anterior, 2.0 mm right lateral, and 3.0 mm ventral to the bregma). A total of 0.075 U collagenase IV dissolved in 4 µl PBS was injected in 10 min using a micro-infusion pump (WPI, Sarasota, FL). The needle was left in place for 5 min to avoid reflux. After with-drawal of the needle, the scalp was sutured. The animals were allowed to recover on a 37 °C heating pad after the operation.

### Tissue collection.
For immunostaining, the mice killed at 14 days poststroke were perfused with PBS followed by 4% paraformaldehyde (PFA, Sinopharm Chemical Reagent, China). Brain samples were removed, fixed in 4% PFA for 2 hrs, and fully dehydrated in 30% sucrose for 2 days at 4 °C, then freeze in −80 °C. Histological cryosections (30 µm in thickness) from anterior commissure to hippocampus were collected. The mice sacrificed at 1 day poststroke were perfused with PBS followed by 4% PFA, the brains were removed, immediately frozen in liquid nitrogen (−20 °C, 10 min), and then freeze in −80 °C. Immunostaining was carried out by sampling four to five sections that collectively spans the entire injury region each mouse.

For western blot, the mice were perfused with PBS, the brain was rapidly removed and placed in a cold mouse brain matrix. The brain was cut into a 2 mm-thick slice before and behind the center of Willis Circle (ischemic brain) or injection site (hemorrhagic brain). A punch (diameter = 2 mm) was used to separate the target region in the ipsilateral striatum (Supp. Fig. 1). Then the tissue was transferred into precooled protein lysis buffer (RIPA with protease cocktail inhibitor, phosphatase inhibitor) to extract protein.

### Immunostaining.
Brain sections incubated with 0.3% TritonX-100 (Sigma, St Louis, MO) and blocked with 1% bovine serum albumin (BSA, Gbico, MA), then incubated with goat anti-Iba-1 (1:200, NB100–1028, Novusbio, CO), goat anti-GFAP (1:400, ab53554, Abcam, CA), rabbit anti-SYP (1:200, ab52636, Abcam), rabbit anti-Homer-1 (1:200, ab184955, Abcam), rat anti-MERTK (1:200, ebio-14-5751-82, eBioscience, CA), rabbit anti-MEGF10 (1:200, ABC10, Millipore, MA) and rat anti-Mac-2 (1:200, CL8942AP, Cedarlane, Canada), rat anti-P2RY12 (1:50, 848002, BioLegend), rat anti-F4/80 (1:50, ab6640, Abcam), rat anti-LAMP2 (1:200, mabc40, Millipore, MA), goat anti-CD31 (1:200, AF3628, R&D), mouse anti-MAP2 (1:200, MAB3418, Millipore, MA), rat anti MBP (1:200, ab-7349, Abcam), mouse anti-SMI32 (1:200, 801701, BioLegend) at 4 °C overnight. After rinsing with PBS for three times, the brain sections were incubated with the secondary anti-bodies: Alexa Fluor 488-conjugated donkey anti-goat, Alexa Fluor 594-conjugated donkey anti-goat, Alexa Fluor 488-conjugated donkey anti-rabbit, or Alexa Fluor 594-conjugated donkey anti-rat, Alexa Fluor 647-conjugated donkey anti-goat, Alexa Fluor 594-conjugated donkey anti-mouse (1:400, Invitrogen, CA), Alexa Fluor 647-conjugated chicken anti-rat (1:400, Invitrogen, CA) for 1 hr at room temperature (RT). For the cell death detection, one step TUNEL Apoptosis Assay Kit (MA0224-2, Meilun, China) was used. Then the brain sections were rinsed with PBS three times and incubated with DAPI (Life Technologies, Mulgrave, VIC, Australia) for 5 min at RT. After rinsing with PBS, the brain sections were covered and sealed with a mounting medium (Vector Labs, Burlingame, CA). Images were acquired using a confocal microscope (Leica, Wetzlar, Germany).

### Fluorescence in situ hybridization.
In situ hybridization was performed using biotin-labeled riboprobes for MEGF10 (5′-CAACCTAACAATTTCATCATTC TGG-3′, 5′-CCTAACAATTTCATCATTCTGGAAT-3′ and 5′-ACTTCCAACT GTCACAACCTAACAA-3′) and MERTK (5′-ACCGTCAGTCCTTTGTCATTGT GGGC-3′, 5′-GGCTAGGGTTGACGAGGGTGCGTAATCT-3′, and 5′-TATGGT GAGACCAGGAGACGCCATTT-3′). Probes were synthesized and subsequent fluorescence in situ hybridization was performed by GenePharma (Shanghai, China). Briefly, brain cryosections (30 µm) were reactivated in citrate buffer (RT, 15 min), treated with proteinase K (37 °C, 20 min) following blocked (37 °C, 30 min), denatured (78 °C, 2 min), dehydrated using graded ethanol (RT, 2 min) and then hybridized with probe working solution (biotin-probe: SA-Cy3: PBS = 2:1:7) in hybridization buffer overnight at 37 °C. Hybridized sections were washed with 0.2× saline-sodium citrate (SSC) buffer at 60 °C for 15 min (wash I), with 2× SSC at 60 °C for 30 min (wash II), and washed at 37 °C for 30 min (wash III). For immunostaining, slices were incubated with anti-Iba-1 (1:200, NB100–1028, Novusbio) and anti-GFAP (1:200, ab53554, Abcam) primary anti-body overnight at 4 °C. After being washed with PBS, slices were incubated with

second antibody (1:400, A11055, Invitrogen). Images were visualized under confocal microscope.

**Western blotting**. Equal amounts of protein (30 μg) were loaded onto 10% (W/V) sodium dodecyl sulfate–polyacrylamide gel electrophoresis and electrophoresed. The proteins were transferred onto PVDF membrane (Millipore) and incubated with the primary antibodies of MEGF10 (1:700, A10508, ABclonal, MA), MERTK (1:700, AF591, R&D, MN), SYP (1:800, ab52636, Abcam), Homer-1 (1:800, ab184955, Abcam) and β-actin (1:1000, MA5–15739, Invitrogen) at 4 °C overnight. The membrane was washed in TBST buffer and incubated with horseradish peroxidase (HRP)-conjugated anti-rabbit or anti-mouse IgG (1:5000, Invitrogen) for 1 hr at RT, and then reacted with an enhanced chemiluminescence substrate (Meilunbio, Shanghai, China). The result of chemiluminescence was recorded and semi-quantified using the ImageJ software (NIH, Bethesda, MD). Bright-field image and chemiluminescent blots are merged using Tanon GIS software (www.Bio-tanon.com.cn).

**qPCR**. Total RNA from the striatum region of brain was isolated using TRIzol Reagent (Invitrogen, Carlsbad, CA). RNA concentration was examined using a spectrophotometer (NanoDrop 1000, Thermo Fisher) followed by a reverse transcription process using cDNA Synthesis SuperMix Kit (11123ES60, Yeason, China). SYBR Green Master Mix (11203ES08, Yeason) was used to perform real-time PCR. A two-stage amplification reaction was performed under the following conditions: 95°C for 5 min, followed by 40 cycles at 95°C for 10 sec, and at 60°C for 30 sec. The primer sequences were:

MEGF10 forward primer: 5′-CCCTCACTGTGCTGATAAATGT-3′, reverse primer: 5′-TGATGGGGTTACACAAAGCTC-3′;

MERTK forward primer: 5′-CCTAACCGTACCTGGTCTGAC-3′, reverse primer: 5′-GGGAGGGGATTACTTTGATGTTG-3′;

GAPDH forward primer: 5′-GAGGGATGCTGCCCTTACC-3′, reverse primer: 5′-AAATCCGTTCACACCGACCT-3′

**Golgi-Cox staining**. For Golgi-Cox staining, the manufacturer's instructions (FD Rapid GolgiStain™ Kit, MD) were strictly followed. Brains were quickly removed, rinsed with double distilled water and then immersed in a 1:1 mixture of solutions A and B (containing mercuric chloride, potassium dichromate and potassium chromate) for 2 weeks at RT in the dark. Brains were then transferred to solution C and kept in the dark for 4 days. Solution C was replaced after the first 24 hrs. Then brains were rapidly frozen in isopentane and kept at −80 °C until sectioning. Cryosectioning was performed on a sliding microtome (Leica) at −22 °C. Coronal sections of 100 μm thickness were cut and transferred to microscope slides (LabScientific) onto small drops of solution C and allowed to dry at RT in the dark overnight. Serial sections were stained with mixture of D and E solutions for 10 min, then dehydrated in a series of graded ethanol, cleared in xylene, and coverslipped with neutral resin. Single-plane images were taken under light field using a confocal microscope (Leica), the following measurement was performed using RECONSTRUCT software (http://synapses.clm.utexas.edu).

**Immuno-TEM**. After transcardial perfusion of buffered 4% PFA, brain tissues were immediately immersed in 0.1 M PB containing 4% PFA and 0.5% glutaraldehyde (pH 7.4) at 4 °C for 6 h, and 200-μm-thick slices were cut with a vibratome (VT-1200S, Leica) and then stained exactly as described in the DAB kit instructions (abs957, Absin, Shanghai, China). The brain slices were incubated with anti-Iba-1 (1:200, WAKO, Tokyo, Japan) and anti-GFAP (1:200, Millipore) primary antibody and secondary antibody conjugated with HRP. Immunoreaction products were visualized using DAB substrate. Following observation under a light microscope and identification of the gliosis region, tissue pieces were cut into 1 mm × 1 mm, additionally fixed with 2.5% glutaraldehyde, and prepared for TEM imaging. In brief, tissues were washed with PB, treated with 1% OsO4 for 90 min at RT. Then tissues were dehydrated in a graded series of ethanol (50%, 70%, 90%), and incubated with 100% acetone, a 1:1 mixture of epoxy resin and acetone, and 100% resin (overnight). Tissues were embedded in epoxy resin and cured in a 60 °C oven for 72 hrs. In all, 1-μm-thick sections were cut and stained with 0.1% toluidine blue to distinguish the target region with a light microscope. 100 nm-ultrathin sections were stained with lead citrate and examined under TEM.

**Cresyl violet staining**. Cresyl violet staining was examined as previously described[61,62]. For measurement of brain atrophy volume or enlarged LV volume, a series of 30 μm in thickness and 300 μm in interval brain cryosections from anterior commissure to hippocampus were collected. For measurement of infarct volume, frozen brain sections 20 μm in thickness and 200 μm in the interval from the anterior commissure to the hippocampus were collected. The sections were stained with 0.1% cresyl violet solution (Meilun, China), and then the brain atrophy, infarct or enlarged LV volume was measured by subtracting the stained area in the ipsilateral hemisphere area from the contralateral hemisphere using ImageJ software (NIH, Bethesda, MD). The brain atrophy volume, infarct volume, or enlarged LV volume were calculated with the following formula: $V = \sum h/3[\Delta S_n + (\Delta S_n * \Delta S_{n+1})^{1/2} + \Delta S_{n+1}]$, in which $V$ represents volume, $h$ represents the

distance between the two adjacent brain sections, $\Delta S_n$ and $\Delta S_{n+1}$ represent the differences between the two adjacent sections.

**IgG staining**. For IgG staining, a series of 20 μm in thickness and 200 μm in interval brain cryosections from the front of hemorrhagic tissue to the end were collected. IgG staining was performed using Histostain™ Plus Kit (SP-0022, Bioss, China). Briefly, brain slices were incubated in 3% $H_2O_2$ for 20 min, and blocked using goat serum for 20 min. Followed by incubating with biotinylated goat anti-mouse IgG solution (1:50, SP-0022, Bioss, China) for 20 min. DAB staining was used for the visualization of immune reactivity (Vector Labs), and the sections were counterstained with hematoxylin (Meilun, China)[62]. The average hemorrhagic area was calculated using ImageJ software.

**Neurobehavioral tests**. Neurobehavioral tests were performed before and 1, 3, 7, and 14 days after MCAO surgery and collagenase-induced hemorrhage by an investigator blinded to the experimental design and treatment.

*Modified neurologic severity score (mNSS)*. The mNSS includes a composite of motor, reflex, and balance tests. The severity score was graded at a scale from 0 to 14, in which 0 represents normal, and a higher score indicates a more severe injury[59].

*Rotarod test*. The rotarod test is used to assess motor coordination and balance alterations. Mice were trained for 3 consecutive days before surgery. The speed was slowly increased from 20 to 40 rpm/min in 5 min. Each mouse was given three trials, and the time that the mice remained on the accelerating rotating rod was recorded.

*Grid-walking test*. The elevated grid-walking apparatus was manufactured using wire mesh with a grid area of 32 cm/20 cm/50 cm (length/width/height). Each mouse was placed individually on the grid and allowed to walk freely for 5 min. A camera was placed beneath the apparatus that allowed recording the animal's walking trace. A step was considered a stepping error (foot fault) if it did not provide support and the foot went through the grid hole. During this 5-min period, the total number of foot faults for each limb, along with the total number of steps, were counted. The foot fault index was calculated by [(contra faults-ipsi faults)/total steps] × 100%.

*Step-through passive avoidance test*. Smart cage used in this test was purchased from AfaSci (SFO, CA) and was used for evaluation of the learning and memory performance with a step-through box consisting of a bright chamber connected to a dark chamber via a door[63]. During the training, once entered into the dark zone, the mouse received an electric foot shock. In all, 24 hrs after the training, the time spent in the dark zone of the mouse was recorded up to 10 min without foot shock. Data were analyzed automatically using the Windows-based program CageScore 2.6.

**Virus production and injection**. AAV (serotype 2/9) was packaged by OBiO Technology Corp., Ltd (Shanghai, China). pAAV-hSyn-PSD95-mCherry-egfp-CW3SL and pAAV-hsyn-Gephyrin-mCherry-eGFP were gifts from Dr. Won-Suk Chung (Korea Advanced Institute of Science and Technology, Daejeon, South Korea). After purification, the viral titer was determined by real-time PCR.

Three weeks before MCAO or collagenase injection, mice were anesthetized using isoflurane and fixed on a stereotaxic frame (RWD, Shenzhen, China). A total volume of 0.4 μl of PBS containing $5 \times 10^9$ viral particles was injected stereotactically at a rate of 0.05 μl/min at 2 mm lateral to the bregma and 3.5 mm under the dura using a micro-infusion pump (WPI, FL). The needle was left in place for 10 min to avoid reflux. After withdrawal to 3.0 mm ventral to the bregma, repeat the steps above. The scalp was then sutured. The animals were allowed to recover on a 37 °C heating pad after the operation.

**Single-cell RNA sequencing**. Brains from healthy mice ($n = 8$), ischemic stroke mice ($n = 8$), or hemorrhagic stroke mice ($n = 8$) were rapidly removed, and target tissues were carefully collected and dissociated using an adult brain dissociation kit from Miltenyi Biotec (Bergisch Gladbach, Germany). Subsequent tissue processing and data acquisition were performed by Oebiotech (Shanghai, China). Briefly, single-cell gel beads in emulsions (GEMs) were generated by loading single-cell suspensions onto a Chromium Single-Cell Controller Instrument (10X Genomics). Approximately 12,000 cells were added to each channel. After that, reverse transcription reactions were engaged to generate barcoded full-length cDNA, and cDNA clean-up was performed with DynaBeads Myone Silane Beads (Thermo Fisher Scientific). Next, cDNA was amplified by PCR and the amplified cDNA was fragmented, end-repaired, A-tailed, and ligated to an index adaptor, and then the library was amplified. Every library was sequenced on a HiSeq X Ten platform (Illumina), and 150 bp paired-end reads were generated. the scRNA-seq analysis was performed in one batch or in several batches.

**Bioinformatic analysis**. The Cell Ranger software pipeline (Version 3.1.0) provided by 10X Genomics was used to demultiplex cellular barcodes, map reads to the genome and transcriptome using the STAR aligner, and down-sample reads as required to generate normalized aggregate data across samples, producing a matrix of gene counts versus cells. We processed the unique molecular identifier (UMI) count matrix using the R package Seurat[64] (Version 3.1.1). To remove the batch effects in single-cell RNA-sequencing data, the mutual nearest neighbors (MNN) was performed with the R package batchelor[65]. To remove low-quality cells and likely multiplet captures, which was a major concern in microdroplet-based experiments, we applied a criterion to filter out cells with UMI/gene numbers out of the limit of mean value ± 2-fold of standard deviations assuming a Gaussian distribution of each cells' UMI/gene numbers. Following visual inspection of the distribution of cells by the fraction of mitochondrial genes expressed, we further discarded low-quality cells where 30% of counts belonged to mitochondrial genes. Library size normalization was performed in Seurat on the filtered matrix to obtain the normalized count.

Top variable genes across single cells were identified using the method described previously[66]. In brief, the average expression and dispersion were calculated for each gene, genes were subsequently placed into several bins based on the expression. Cells were clustered based on a graph-based clustering approach, and were visualized in two-dimension using tSNE. Likelihood ratio test that simultaneously test for changes in mean expression and in the percentage of expressed cells was used to identify significantly DEGs between clusters. Here, we use the R package SingleR[67], a computational method for unbiased cell type recognition of scRNA-seq to infer the cell of origin of each of the single cells independently and identify cell types. DEGs were identified using the Seurat[64] package. $p < 0.05$ and $|log2foldchange| > 0.58$ was set as the threshold for significantly differential expression. GO enrichment and KEGG pathway enrichment analysis of DEGs were respectively performed using R based on the hypergeometric distribution.

**Image acquirement**. Quantitative analysis of the acquired images was performed using both LAS AF Lite (for quantification of cell numbers), ImageJ software (for quantification of fluorescence intensity), and Imaris (for 3D rendering construction of analyzed cells) software. Images at four perifocal areas surrounding lesion core in the ipsilateral hemisphere were taken (Experimental scheme) under a confocal microscope. For Mac-2 and MEGF10/MERTK expression, ×40 and ×160 images were taken, z-stack = 5, 5 μm per section; for synaptic engulfment and colocalization, ×40 and ×200 image were taken, z-stack = 10, 2.5 μm per section, the 3D view images were reconstructed in Imaris software (9.0.3 Bitplane).

The percent volume of engulfed synaptic material in glial cells was assessed using ImageJ software and DIANA plugin as previously described[23,68] with modification. In brief, since each image was taken under the same conditions, we introduced a normalization value to consider the differences in synaptic puncta volume. Synaptic puncta in-unit glia represents the size of synaptic puncta inside of glia/ the size of total phagocytic glia. mCherry-alone puncta in-unit glia (AU) represents the size of mean mCherry-alone puncta inside of unit glia = the size of mCherry-alone puncta inside of glia/ the size of total phagocytic glia. The size of mCherry-alone puncta inside(outside) of glia = the size of mCherry-GFP puncta inside(outside) of glia–the size of GFP puncta inside(outside) of glia. The percentage of synaptic puncta inside of glial lysosome was assessed using ROI quantification tool of LAS AF Lite software. The average percentage of synaptic puncta within lysosome inside of glia = (ROI of synaptic puncta inside of glial lysosome/ROI of total synaptic puncta inside glia)×100%. For pre- and postsynaptic puncta colocalization assay, the number of synaptic puncta per field was measured using DIANA plugin as described[23]. For other colocalization assays, the number of positive cells was calculated in LAS AF Lite software. For dendritic spine quantification, representative brain coronal sections (100 μm-thick) from lesion site were imaged under confocal microscope, ×20 and ×120 images were taken at single panel. For synaptic ultrastructure observation, 100 nm-ultrathin section was imaged under biological TEM (Tecnai G2 spirit Biotwin) at ×2.9 K, ×9.3 K, ×30 K and ×48 K, 30–40 DAB-stained cells were quantified.

**Data analysis and statistics**. All statistical tests were run in Prism7.0 (Graphpad). All values are presented as mean ± SD. The number of brain sections and animals used (*n*) is indicated in the figure legends. Statistical analyses were performed with one-way or two-way analysis of variance, followed by Dunnett or Tukey multiple comparisons. Two-tailed $p < 0.05$ were considered statistically significant.

**Reporting summary**. Further information on research design is available in the Nature Research Reporting Summary linked to this article.

## Data availability
The scRNA-seq data generated in this study have been deposited in the Gene Expression Omnibus under the accession number GSE167593. Uncropped western blots are provided in the Supplementary Information file (Supp. Fig. 12). Source data are provided with this paper.

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

## Acknowledgements
This study was supported by grants from the National Key R&D Program of China #2016YFC1300602 (G.Y.Y.), #2019YFA0112000 (Y.T.), the National Natural Science Foundation of China (NSFC) projects 81771251 (G.Y.Y.), 81801170 (Y.T.), 82071284 (Y.T.), 81771244 (Z.Z.), 81974179 (Z.Z.), 81870921 (Y.W.), the Scientific Research and Innovation Program of Shanghai Education Commission 2019-01-07-00-02-E00064 (G.Y.Y.), Scientific and Technological Innovation Act Program of Shanghai Science and Technology Commission, 20JC1411900 (G.Y.Y.), the National Research Foundation of Korea (NRF) grant (2020M3E5D9079912, 2021R1A2C3005704, W.S.C.), the Korea Health Technology R&D Project (HU20C0290, W.S.C.) and K.C. Wong Education Foundation (G.Y.Y.). We would like to thank OE Biotech Company (Shanghai, China, http://www.oebiotech.com/) for providing scRNA-seq, and thank Dr. Qidong Zu and Dr. Yongbing Ba for the assistance with bioinformatics analysis.

## Author contributions
G.Y.Y., Y.T., and W.S.C. conceived the project, designed the experiments, and edited the paper finally. X.S. and L.L. designed and performed the experiments, analyzed the data, and drafted the manuscript and figures. J.W. participated in scRNA-seq, FISH, and data analysis. H.S. contributed to behavior tests and immunostaining. Y.L. helped with sample collection. M.M. contributed to the western blot. C.L., R.S., and J.H.L. contributed to animal breeding and identification. H.T., Z.Z., and Y.W. helped to design the experiment and interpret the data.

## Competing interests
The authors declare no competing interests.
