## [Peer Review File · Nature Communications]

Reviewers' Comments:

Reviewer #1:

Remarks to the Author:

The manuscript by Shi, Lou, and Wang et al. seeks to investigate whether microglia/macrophage and astrocytes have a different phagocytic role in ischemic and hemorrhagic stroke. The authors first provide evidence that while astrocytes and microglia both appear to engulf synaptic material in ischemic stroke, only microglia engulf material in hemorrhagic stroke. They then use transgenic mice in which MEGF10 or MERTK phagocytic signaling is disrupted in either microglia/macrophages or astrocytes and show that disruption in either cell type attenuates brain damage and improves behavioral outcomes in the ischemic stroke model. In contrast, only the ablation of MEGF10 and MERTK from microglia/macrophages, but not astrocytes, showed a comparable protective effect following hemorrhagic stroke. This provides interesting new evidence that astrocyte responses and phagocytosis are differentially regulated in different stroke models and a new role for MEGF10 and MERTK in stroke. Unfortunately, there are significant concerns regarding the lack of quantification and controls as well as lack of data to sufficiently support some conclusions. These and other concerns are outlined below:

Major points:

1. SHAM controls are lacking. The authors only included data for SHAM-treated mice in Figure 4. Therefore, it is unclear what the baseline of the assessed parameters is in healthy controls. Related to this point, it is unclear which treatment group the SHAM-treated mice in Figure 4 correspond to and why only one SHAM group was included. Every time point and model should have its own SHAM control to increase rigor.
2. Many data lack quantification. This includes the data shown in Figure 1c-d and g-h (quantification of engulfment), Figure 2 (quantification of signals inside and outside of microglia/macrophages and astrocytes), Figure 3 (quantification of engulfment), Figure 4b-c and e-f (quantification of MERTK/MEGF10 positive and negative cell numbers), Figure 6 (quantification of synaptic inclusions), and Supplement Figure 2 e-h (quantification of MERTK/MEGF10 positive and negative cell numbers). To make any interpretation and conclusions, data should be quantified and statistical analyses should be performed.
3. The authors correctly cite previous work in the introduction describing that astrocytes use MEGF10 and MERTK phagocytic pathways while microglia/macrophages preferentially use the classical complement pathway to mediate synapse elimination. Was there a particularly reason why MEGF10 and MERTK, but not complement, were chosen for this study? Given the central role of complement signaling for microglial synapse engulfment in health and disease, it is important to consider.
4. In Figure 4a,d authors show that MEGF10 and MERTK protein levels are both significantly upregulated 14 days after the induction of ischemic or hemorrhagic stroke but not changed on days 1-7. In contrast, the authors show an upregulation of Mac-2 (a phagocytic marker of microglia and astrocytes) already 1 day following stroke induction in both models (Fig. 1a,3). How do authors explain this discrepancy in the timing of different phagocytic markers? Are synapses already engulfed during early stages (d1 post induction)? If yes, would MERTK/MEGF10-independent mechanisms underly early synapse removal? Perhaps, complement?
5. When studying stroke pathology, it is critical to distinguish if the studied area is an area with hemorrhage or without, or if it is within the core vs. the penumbra. Authors need to provide a better description of the analyzed brain regions and make sure that the areas are comparable across different experiments. It would also be helpful to show an image of a larger area and show exactly where the analyzes were performed.
6. To assess microglia/macrophages, the authors assessed cells using the marker Iba1. They correctly call these cells microglia/macrophages in the text, but sometimes refer to areas with more reactive cells as microgliosis. It would be most informative, therapeutically relevant, and impactful to identify which cell type is engulfing using microglia-specific markers as (e.g., P2RY12, Clec7a, etc.) to distinguish microglia from infiltrating macrophages.
7. The size of synaptic puncta within the cells appears to differ largely (e.g., see Figure 1g, microglia cell with one giant and several small puncta). Are these puncta counted as 1 puncta regardless of size? How do authors ensure that the large puncta (presumed to count as 1 puncta) are not comprised of many synapses? It would be more informative to measure the volume of engulfed synaptic material within the cells, as this assessment also comprises the current gold standard in the field.

8. Similar to the point above, there lacks any description of how images were acquired and analyzed in the paper's methods section. This is critical to judge the quality of the data. Were the data acquired as z-stacks? If so, how many z-stacks, what thickness, what magnification. How many fields of view were analyzed per animal and how were they chosen? How exactly was engulfed material defined and quantified?
9. The authors should quantify astrocyte engulfment for the hemorrhagic mice in Figures 2 and 3 as this contrast to ischemic stroke is central to the manuscript.
10. A more extensive validation of the MERTK and MEGF10 conditional knockouts should be performed. This should include the downregulation of mRNA levels and the inclusion of oil-injected control mice to rule out spontaneous recombination. Also, more rigorous quantification of immunofluorescence data in microglia, astrocytes, and other cell types that express MERTK should be included.
11. The orthogonal projections in Figures 1-3 are not very clear and relatively small that it is difficult to appreciate the engulfment of synaptic material within the cells. A better way to show this is to provide 3D renderings of the analyzed cells.
12. In Figures 1-3, authors show increased engulfment of synaptic material following ischemic and hemorrhagic stroke. However, it remains unclear if these increases in synaptic engulfment lead to an overall reduced synapse density. The western blots are inconsistent if you compare the banding pattern across blots. For instance, SYP levels appear very different between the two MEGF10 WT mice shown in Fig. 5. This raises concerns about the reproducibility of these results. Also, this type of quantitative analysis only quantifies total protein vs. structural synapses. It would be better to quantify puncta number and colocalized pre and post-synaptic markers by immunofluorescence confocal microscopy.
13. In the description of figure 2 authors emphasize that red puncta within microglia and astrocytes lost the green fluorescence to prove that their AAV paradigm is working. However, there are numerous GFP-negative puncta localized outside the labeled cells? In addition, there are also several bright green puncta detected within the cells? Are those signals not yet localized to lysosomes? If not, what subcellular compartment do they localize to?
14. The data in figure 8 and figure 9 assess brain atrophy and ventricle enlargement in both genotypes and models 14 days after the stroke. Were there any differences in the infarct volume 24h after the stroke between genotypes? A TTC staining, Nissl staining, or MRI imaging should be shown to prove that the initial infarct volume is the same across the different transgenic mice. The initial infarct volume could drastically influence the outcome and the development of the disease at later timepoints. Also, a quantification of the hemorrhagic area by DAB staining should be provided. There is an evident difference between WT and C-MEGF10KO and A-MEGF10KO hemorrhagic area that could be affecting the whole process in the acute timepoints.
15. In figure 7 and 8, the authors show that MEGF10 and MERTK cKO mice have reduced brain atrophy. As both stroke models have been described to cause severe cellular damage and induce substantial cell death, does this reduction in brain atrophy purely reflect protection of synapses? Or do these genetic manipulations also affect cell death and/or axonal degeneration independent of protecting synapses?
16. Are MEGF10 and MERTK detected in their single cell RNA seq analysis? Given the central role of both genes this data should be included into the figures 10 and S3.
17. Sup Fig.4 shows the expression of MEGF10 and MERTK in hemorrhagic post mortem brain tissue; however, no control tissue was assessed. This data should be described in the result section and compared to controls to better interpret this result.
18. For scRNA-seq experiments, the authors state that 12,000 total cells were analyzed per condition, how many astrocytes and microglia were analyzed? As the number of cells analyzed is the best predictor of the quality of scRNA-seq data sets, this information should be included for each cell type.
19. In the scRNA-seq, were biological replicates pooled? Or were samples from each biological replicate run separately? If the latter, the authors should show the distribution of the cell types across biological samples to ensure equal distribution of the bio replicates on the tSNE plot.
20. For the scRNA-seq, why were no neuron or oligodendrocyte precursor cell clusters detected?
21. Given the work of Liddel et al. Nature 2017, it would be informative to compare the gene expression profiles of the astrocytes in the different ischemia models to the A1 vs. A2 astrocytes. If the astrocytes in the hemorrhagic stroke more closely resemble the A1 cells, this would be consistent with this previous study that showed that A1 astrocytes are less efficient at phagocytosing cellular material and could explain why there are differences in astrocyte

engulfment between the two models.

Minor points:

1. The graphical abstract does not summarize the findings of the paper well. A better representation should be selected.
2. The authors evaluate the neurological outcome in the ischemic stroke model using four different tests (mNSS, Rotarod test, Grid walking test, and Step-through passive avoidance test). It is unclear why in the hemorrhagic model the step-through passive avoidance test was not performed?
3. In line 166, the authors say that MEGF10 and MERTK were elevated 7 and 14 days after stroke. Figure 4d, however, does not show a significant increase at day 7.
4. In figure 5, the representation of the statistical significance is not clear. When comparing two different groups, the symbol should be different, or lines indicating which two groups are being compared should be added for clarity.
5. In Figure 9c, the statistical analysis showed in the graphs does not match the one described in the text. Fig 9c is described in the text as "MEGF10KO in astrocytes showed no effects on vascular enlargement" however, in the figure, there is a # between MEGF10WT and A-MEGF10KO indicating a statistical difference of $p < 0.05$. This should be corrected.
6. Figure 5j and 5l show that the number of engulfed puncta in microglia/macrophages is very similar to the number in astrocytes. However, the images shown in figure 1g and the text's description suggest much more puncta are engulfed by microglia after hemorrhagic stroke compared to astrocytes.
7. In Figures 1 and 4 authors only show merged images in what appear to be maximum intensity projections. Since it is difficult to appreciate the individual channels and co-localization, the authors should also show the individual channels and single imaging planes.
8. Line 107: Fig. 1e should be Fig. 1g. Same for line 114.
9. Line 111: It describes Fig. 1c, but it is not specified in the text. It should be included for clarity.
10. The manuscript should be edited for grammar. There are numerous instances of grammatical errors throughout.

Reviewer #2:

Remarks to the Author:

This is an elegantly presented and comprehensive study of microglial and astrocytic involvement in synapse elimination in ischemic and haemorrhagic stroke. Synapse engulfment and phagocytosis by these cells have been shown recently, however, the comparison between the two pathological events using advanced tools and showing the beneficial effects of blocking phagocytosis provide advances in the field. In particular, it is interesting that astrocytes seem to have less direct contribution to synapse phagocytosis in haemorrhagic stroke. The systematic approach using immunohistochemical, biochemical, transcriptional and behavioural analyses and the tools developed by the authors, including synapse reporters, and various conditional knock-out mice have resulted in confidence in the findings. I have relatively minor comments, which should be addressed by the authors.

Minor comments:

1. The EM images in Fig. 6 are low resolution and the blue overlays are obscuring the demonstration of synaptic inclusions (SI) in the cells. Some of these would need replacement with higher quality/higher resolution images in which the synaptic vesicle can be clearly visualised in the SI.
2. The authors should specify in the methods if only one-way ANOVA or two-way ANOVA tests were also used in the study. In addition, the type of statistical tests used should be indicated in the figure legends.
3. It should also be specified in the methods if the scRNA-seq analysis was performed in one batch or in several batches. If the latter, was any computational batch correction used? They should also comment on whether any filtering was carried out for cells with disproportionately high

mitochondrial genes (possibly dead cells).

4. A heatmap for unbiased clustering and also for cell-type specific DEGS should be included, representing the cell-type identities and their marker genes. This is to better demonstrate how cell populations were identified in the datasets, and whether this was consistent between the two experimental groups.

5. It is not indicated in any of the scRNA-seq-related figures whether the microglial and astrocytic expression of MEGF10 and MERTK was altered in the dataset. If this was not the case, then it should be discussed how this relates to the protein level findings (by WB) in the main part of the manuscript.

6. The difference in astrocytic phenotype/function between the ischemic and hemorrhagic stroke is interesting. The potential causes of this should be also discussed more explicitly in the manuscript. In specific, it would be pertinent to include if there is any evidence in their scRNA-seq data for a different inflammatory/immune cell composition in the two lesion types, which may have had a differential impact on astrocytes.

Reviewer #3:

Remarks to the Author:

The interesting study by Shi et al. focuses on the role of reactive gliosis in stroke and in cellular and functional recovery in ischemic and hemorrhagic stroke. The authors report some exciting findings regarding 1) molecular and functional differences in microglia and astrocytes in the glial scar in these two different types of strokes, and 2) the distinct roles that these cell types play in the extent of stroke-induced injury, and in regeneration and functional recovery. The authors specifically delete MEGF10 and MERTK phagocytic receptors in microglia and astrocytes to inhibit phagocytosis in these distinct cell types. In hemorrhagic stroke, inhibiting phagocytosis in microglia/macrophages, but not in astrocytes, improved cellular, anatomical and functional outcomes. Conversely, in ischemic stroke, inhibiting phagocytosis in microglia/macrophages and in astrocytes, improved brain damage and functional outcomes. The authors conclude that reactive microgliosis and astrogliosis play individual roles in mediating synapse engulfment in different stroke models, and preventing the individual roles of these cells can rescue synapse loss and differentially attenuate the consequences of stroke.

This is an important and novel report, as it demonstrates not only differences in reactive gliosis in different stroke models, but also the impact of phagocytosis in distinct cell types during the recovery process. These findings could eventually lead to designing more targeted therapies in different types of stroke aimed at attenuating brain injury and improving functional recovery.

I would recommend some revisions, before considering this article for publication.

1. Figure 4. Western blot in Panel d. the levels of MEGF10 dramatically decrease at D3, between D1 and D7. However, the quantification does not show that. The Western blot should match the quantification and vice-versa.
2. Figure 6. This figure shows that conditional MEGF10 and MERTK deletion reduces glial-cell mediated synapse engulfment after stroke, and that there are differential effects in hemorrhagic vs. ischemic stroke. However, this figure is not quantitative, and quantification of engulfed synaptic elements should be performed to convincingly support the morphological data.
3. Page 9. Figure 7. In hemorrhagic stroke, MEGF10 and MERTK deletion only slightly increased the number of total dendritic spines and mature spines in the gliosis region, as shown in Figure 7h-l. The difference is much larger in ischemic stroke. This should be reworded in the text.
4. The authors convincingly demonstrate differences in brain atrophy in distinct types of stroke after MEGF10 and MERTK deletion. It would be important to also characterize changes in white matter, and how these changes affect brain atrophy in different KO mouse lines in distinct stroke models.
5. Figures 8 and 9. I am not sure about the rationale for organizing these two figures in the current layout. I would recommend a different layout by combining brain atrophy/ventricular

enlargement data in one figure (Figure 8) and all behavioral data in a separate figure (Figure 9).

6. The grammar and English of the whole manuscript need significant revisions.

Response to comments of the reviewers

Reviewer #1 (Remarks to the Author):

The manuscript by Shi, Lou, and Wang et al. seeks to investigate whether microglia/macrophage and astrocytes have a different phagocytic role in ischemic and hemorrhagic stroke. The authors first provide evidence that while astrocytes and microglia both appear to engulf synaptic material in ischemic stroke, only microglia engulf material in hemorrhagic stroke. They then use transgenic mice in which MEGF10 or MERTK phagocytic signaling is disrupted in either microglia/macrophages or astrocytes and show that disruption in either cell type attenuates brain damage and improves behavioral outcomes in the ischemic stroke model. In contrast, only the ablation of MEGF10 and MERTK from microglia/macrophages, but not astrocytes, showed a comparable protective effect following hemorrhagic stroke. This provides interesting new evidence that astrocyte responses and phagocytosis are differentially regulated in different stroke models and a new role for MEGF10 and MERTK in stroke. Unfortunately, there are significant concerns regarding the lack of quantification and controls as well as lack of data to sufficiently support some conclusions. These and other concerns are outlined below:

Major points:

1. SHAM controls are lacking. The authors only included data for SHAM-treated mice in Figure 4. Therefore, it is unclear what the baseline of the assessed parameters is in healthy controls. Related to this point, it is unclear which treatment group the SHAM-treated mice in Figure 4 correspond to and why only one SHAM group was included. Every time point and model should have its own SHAM control to increase rigor.

Response: The reviewers' suggestions are thoughtful and helpful. We added healthy control and sham groups at each timepoint in revised **Fig. 4a, k**. Brain tissues were collected in the same way at 1, 3, 7 and 14 days of sham groups. We re-analyzed the MEGF10 and MERTK protein levels in both sham and stroke groups at 1, 3, 7 and 14 days, and normalized the expression of MEGF10 and MERTK in stroke mice to their corresponding sham groups. We found that the expression of MEGF10 and MERTK were increased at 14 days in both ischemic (**Fig. 4a, b**) and hemorrhagic stroke mice (**Fig. 4k, l**).

2. Many data lack quantification. This includes the data shown in Figure 1c-d and g-h (quantification of engulfment), Figure 2 (quantification of signals inside and outside of microglia/macrophages and astrocytes), Figure 3 (quantification of engulfment), Figure 4b-c and e-f (quantification of MERTK/MEGF10 positive and negative cell numbers), Figure 6 (quantification of synaptic inclusions), and Supplement Figure 2 e-h (quantification of MERTK/MEGF10 positive and negative cell numbers). To make any interpretation and conclusions, data should be quantified and statistical analyses should be performed.

Response: Thanks for the comment. We performed statistical analysis of above data and added corresponding statistical graph in the revised manuscript. We quantified the fluorescent volume of synaptic material and the number of synaptic inclusions within glial cells in revised **Fig. 1;**

we quantified virus signals inside and outside of glial cells in revised **Fig. 2**; we quantified the percentage of synapse within lysosome in revised **Fig. 3**; we quantified MEGF10/MERTK positive cell numbers (number of negative%=100% - number of negative%) in revised **Fig. 4**; we quantified the number of synaptic inclusions within glial cells in revised **Fig. 6**; and quantified MEGF10/MERTK positive cell numbers (number of negative%=100% - number of negative%) in revised **Supp Fig. 4**, respectively. Detailed descriptions were added in the result section.

3. The authors correctly cite previous work in the introduction describing that astrocytes use MEGF10 and MERTK phagocytic pathways while microglia/macrophages preferentially use the classical complement pathway to mediate synapse elimination. Was there a particularly reason why MEGF10 and MERTK, but not complement, were chosen for this study? Given the central role of complement signaling for microglial synapse engulfment in health and disease, it is important to consider.

Response: We agree with the reviewer that complement signaling plays a central role for microglial mediated synapse engulfment in health and diseased brain, which has been well documented in previous studies, including ischemic stroke. For example, complement activation triggered microglial phagocytosis of live neurons in mice 1 day following stroke¹, later in the year of 2020, their group showed complement activation of hippocampal synapses directed microglia-dependent phagocytosis of synapses for at least 30 days after ischemic stroke, leading to a loss of synaptic density². Our lab is very interested in exploring other molecules other than complement that participates in synapse engulfment in stroke models. We previously identified MEGF10 and MERTK as phagocytic molecules that mediate glial cell induced synapse elimination in developing and adult brain^{3,4}, however, whether MEGF10 and MERTK are involved in glial cell induced synapse engulfment after stroke is largely unknown. Here, our study focuses on investigating the effect of MEGF10 and MERTK in synapse engulfment; specifically, in the chronic stage of both ischemic stroke and hemorrhagic stroke, which has never been explored before. The results of our study highlighted that in addition to complement signaling, MEGF10 and MERTK are also important molecules that mediate glial cell-induced synapse engulfment at the chronic stage of ischemic and hemorrhagic stroke. We believe our study will broaden the knowledge in this field.

4. In Figure 4a, d authors show that MEGF10 and MERTK protein levels are both significantly upregulated 14 days after the induction of ischemic or hemorrhagic stroke but not changed on days 1-7. In contrast, the authors show an upregulation of Mac-2 (a phagocytic marker of microglia and astrocytes) already 1 day following stroke induction in both models (Fig. 1a,3). How do authors explain this discrepancy in the timing of different phagocytic markers? Are synapses already engulfed during early stages (d1 post induction)? If yes, would MERTK/MEGF10-independent mechanisms underly early synapse removal? Perhaps, complement?

Response: This is a good question. In our study we found that Mac-2 was upregulated as early as 1 day following stroke, but without changes of MEGF10 and MERTK protein levels. We also found that synapses were engulfed by microglia/macrophages and astrocytes at 1 day after stroke (**Supp Fig. 7**), suggesting that MEGF10 and MERTK did not play critical roles in

mediating early synapse removal following stroke. We agree with the reviewer's opinion that complement component-dependent mechanisms are responsible for early synapse engulfment following stroke, which is supported by previous study. For example, Alawieh et al showed that the complement system is activated by ischemic stroke, and specific local complement inhibition reduced cell death and inflammation induced by stroke¹. This point was added to our discussion.

5. When studying stroke pathology, it is critical to distinguish if the studied area is an area with hemorrhage or without, or if it is within the core vs. the penumbra. Authors need to provide a better description of the analyzed brain regions and make sure that the areas are comparable across different experiments. It would also be helpful to show an image of a larger area and show exactly where the analyzes were performed.

Response: As requested, in the revised **Experimental Scheme**, we added the photographs illustrating the brain regions collected for western blot and images for immunostaining analysis. Detailed description was added in the revised method section.

For immunostaining, the mice sacrificed at 14 days following stroke were perfused with PBS followed by 4% paraformaldehyde (PFA, Sinopharm Chemical Reagent, China). Brain samples were removed, fixed in 4% PFA for 2 hrs, and fully dehydrated in 30% sucrose for 2 days at 4°C, then freeze in -80°C. Histological cryosections (30 μm in thickness) from anterior commissure to hippocampus were collected. The mice sacrificed at 1 day following stroke were perfused with PBS followed by 4% paraformaldehyde, the brains were removed, immediately frozen in liquid nitrogen (-20°C, 10min), and then freeze in -80°C. Immunostaining was carried out by sampling 4 to 5 sections that collectively spans the entire injury region each mouse.

For western blot, the mice were perfused with PBS, the whole brain was rapidly removed and placed in a cold mouse brain matrix. The brain was cut into a 2 mm-thick slice before and behind the center of Willis Circle (ischemic brain) or injection site (hemorrhagic brain). A punch (diameter=2 mm) was used to separate the target region in the ipsilateral striatum (**Experimental Scheme**). Then the tissue was transferred into precooled protein lysis buffer (RIPA with protease cocktail inhibitor, phosphatase inhibitor) to extract protein.

6. To assess microglia/macrophages, the authors assessed cells using the marker Iba1. They correctly call these cells microglia/macrophages in the text, but sometimes refer to areas with more reactive cells as microgliosis. It would be most informative, therapeutically relevant, and impactful to identify which cell type is engulfing using microglia-specific markers as (e.g., P2RY12, Clec7a, etc.) to distinguish microglia from infiltrating macrophages.

Response: To distinguish the engulfing contribution of resident microglia and infiltrated macrophages, we used microglia-specific marker P2RY12⁵ and macrophage-specific marker F4/80⁶, co-labeled with synaptic markers in the revised **Supp Fig. 1**. The proportion of phagocytic microglia (P2RY12⁺/SYP⁺ cells /P2RY12⁺ cells×100%) and phagocytic macrophages (F4/80⁺/SYP⁺ cells /F4/80⁺ cells×100%) was quantified. We found that in ischemic stroke, microglia and macrophages contributed similarly (50%-70%) to engulfing synapse, whereas in hemorrhagic stroke, macrophages (≈80%) contributed more to engulfing synapse than that in microglia (≈50%).

7. The size of synaptic puncta within the cells appears to differ largely (e.g., see Figure 1g, microglia cell with one giant and several small puncta). Are these puncta counted as 1 puncta regardless of size? How do authors ensure that the large puncta (presumed to count as 1 puncta) are not comprise of many synapses? It would be more informative to measure the volume of engulfed synaptic material within the cells, as this assessment also comprises the current gold standard in the field.

Response: Thanks for the suggestion. The percent volume of engulfed synaptic material in glial cells was assessed using *ImageJ* software and DIANA plugin as previous described with modification^{4,7}. Briefly, since each image was taken under the same conditions, we introduced a normalized value to consider the differences in synaptic puncta volume. Synaptic puncta in unit glia represents the size of synaptic puncta inside of glia/ the size of total phagocytic glia. mCherry-alone puncta in unit glia (AU) represents the size of mean mCherry-alone puncta inside of unit glia= the size of mCherry-alone puncta inside of glia)/ the size of total phagocytic glia. The size of mCherry-alone puncta inside(outside) of glia= the size of mCherry-GFP puncta inside(outside) of glia - the size of GFP puncta inside(outside) of glia.

8. Similar to the point above, there lacks any description of how images were acquired and analyzed in the paper methods section. This is critical to judge the quality of the data. Were the data acquired as z-stacks? If so, how many z-stacks, what thickness, what magnification. How many fields of view were analyzed per animal and how were they chosen? How exactly was engulfed material defined and quantified?

Response: We clarified the parameters used to quantify image analysis in revised methods section. Quantitative analysis of the acquired images was performed using both LAS AF Lite (for quantification of cell numbers), *ImageJ* (for quantification of fluorescence intensity) and Imaris (for 3D rendering construction of analyzed cells) software. Images at four perifocal areas surrounding lesion core in the ipsilateral hemisphere were taken (**Experimental Scheme**) under confocal microscope. For Mac-2 and MEGF10/MERTK expression, 40X and 160X images were taken, z-stack=5, 5 μm per section; for synaptic engulfment and colocalization, 40X and 200X image were taken, z-stack=10, 2.5 μm per section, the 3D view images were reconstructed in Imaris software (9.0.3 Bitplane). The percent volume of engulfed synaptic material in glial cells was assessed using *ImageJ* software and DIANA plugin as previous described with modification^{4,7}. Briefly, since each image was taken under the same conditions with the same thickness and intensity, we introduced a normalization value to consider the differences in synaptic puncta volume. Synaptic puncta in unit glia represents the size of synaptic puncta inside of glia/ the size of total phagocytic glia. mCherry-alone puncta in unit glia (AU) represents the size of mean mCherry-alone puncta inside of unit glia= the size of mCherry-alone puncta inside of glia)/ the size of total phagocytic glia. The size of mCherry-alone puncta inside(outside) of glia= the size of mCherry-GFP puncta inside(outside) of glia - the size of GFP puncta inside(outside) of glia. The percentage of synaptic puncta inside of glial lysosome was assessed using ROI quantification tool of LAS AF Lite software. The average percentage of synaptic puncta within lysosome inside of glia= (ROI of synaptic puncta inside of glial lysosome / ROI of total synaptic puncta within glia) $\times 100\%$. For pre- and post- synaptic puncta co-localization assay, the number of synaptic puncta per field was measured using DIANA plugin as described⁷. For other co-localization assay, the number of positive cells was calculated

in LAS AF Lite software. For dendritic spine quantification, representative brain coronal sections (100 µm-thick) from lesion site were imaged under confocal microscope, 20X and 120X images were taken at single panel. For synaptic ultrastructure observation, 1 mm X 1 mm X 100 nm ultrathin section was imaged under biological transmission electron microscope (Tecnai G2 spirit Biotwin) at 2.9, 9.3, 30 and 48K, 30 to 40 DAB-stained cells were quantified.

9. The authors should quantify astrocyte engulfment for the hemorrhagic mice in Figures 2 and 3 as this contrast to ischemic stroke is central to the manuscript.

Response: We added the additional data in the revised **Fig. 2** and **Fig. 3**, accordingly.

10. A more extensive validation of the MERTK and MEGF10 conditional knockouts should be performed. This should include the downregulation of mRNA levels and the inclusion of oil-injected control (transgenic?) mice to rule out spontaneous recombination. Also, more rigorous quantification of immunofluorescence data in microglia, astrocytes, and other cell types that express MERTK should be included.

Response: To address the reviewer's concern, both wild type mice (C57BL6) and transgenic mice were randomly allocated into two groups, intraperitoneally injected with oil (150 µl, once a day for five consecutive days) or tamoxifen (10 mg/ml, 0.075 mg/g body weight, once a day for five consecutive days). mRNA of striatum was evaluated by qPCR. The qPCR results were provided in **Supp Fig. 4b**, which showed that *MEGF10/MERTK* mRNA levels were significantly reduced in the tamoxifen-treated transgenic mice compared to the tamoxifen-treated WT mice, oil-treated WT mice and oil-treated transgenic mice. Thus, we concluded that there is no spontaneous recombination in the transgenic mice we used in our study. Based on the reviewer's suggestion, we re-validated the MERTK and MEGF10 conditional knockouts specificity in the brain. We found that only astrocyte- or microglia-specific *MEGF10* and *MERTK* flox groups with tamoxifen injections exhibited the significant reduction in *MEGF10* and *MERTK* expression, compared to control groups. In addition, as you can see in the **Supp Fig. 4a**, AAV5-GFAP-Cre virus injected into *MEGF10^{fl/fl}* or *MERTK^{fl/fl}* mouse brain showed astrocyte-specific deletion of MEGF10 and MERTK proteins, compared to control virus injected group, further proving the specificity of our floxed lines.

We also examined the MEGF10/MERTK expression in neurons (MAP2⁺) and endothelial cells (CD31⁺) (**Supp Fig. 3**), and reported statistical analysis results in **Fig. 4**. The immunostaining showed MEGF10/MERTK was rarely expressed in MAP2⁺ neurons or CD31⁺ endothelial cells in the striatum.

11. The orthogonal projections in Figures 1-3 are not very clear and relatively small that it is difficult to appreciate the engulfment of synaptic material within the cells. A better way to show this is to provide 3D renderings of the analyzed cells.

Response: To illustrate the intracellular synapse engulfment more clearly, we provided 3D images using Imaris Bitplane 9.0.3 in revised **Figs. 1-3**.

12. In Figures 1-3, authors show increased engulfment of synaptic material following ischemic and hemorrhagic stroke. However, it remains unclear if these increases in synaptic engulfment leads to an overall reduced synapse density. The western blots are inconsistent if you compare

the banding pattern across blots. For instance, SYP levels appear very different between the two MEGF10 WT mice shown in Fig. 5. This raises concerns about the reproducibility of these results. Also, this type of quantitative analysis only quantifies total protein vs. structural synapses. It would be better to quantify puncta number and colocalized pre- and post-synaptic markers by immunofluorescence confocal microscopy.

Response: To address the reviewer's concern, we reperformed Western blot, which was represented in revised **Fig. 5**. To evaluate the overall synapse density, we colabeled pre- and postsynaptic markers and measured the density of SYP⁺ presynapses, PSD95⁺ postsynapses, and SYP⁺/PSD95⁺ synapses, to show the overall synapse density. The results showed that the density of SYP⁺ presynapse, PSD⁺ postsynapse and SYP⁺/PSD95⁺ synapse was increased in C-MEGF10^{KO}, A-MEGF10^{KO}, C-MERTK^{KO} and A-MERTK^{KO} mice after ischemic stroke (**Fig. 5i-k**), as compared to the control. In hemorrhagic mice, the synapse density was increased in C-MEGF10^{KO} and C-MERTK^{KO} mice (**Fig. 5t-v**).

13. In the description of figure 2 authors emphasize that red puncta within microglia and astrocytes lost the green fluorescence to prove that their AAV paradigm is working. However, there are numerous GFP-negative puncta localized outside the labeled cells? In addition, there are also several bright green puncta detected within the cells? Are those signals not yet localized to lysosomes? If not, what subcellular compartment do they localize to?

Response: We appreciate the referee's constructive points, and these points are well-described in our previous publication⁴. In the corresponding images, GFP-negative mCherry puncta outside the labeled glial cells can be originated from two different sources. First, since we labeled microglia and astrocytes separately in **Fig. 2**, most of mCherry puncta outside the labeled cells are likely located in non-labeled microglia or astrocytes. Second, in our previous publication⁴, we also reported that minor population of mCherry alone puncta can be found inside of neuronal processes as well, since synaptic vesicles can be internalized/recycled by neuronal organelles.

Previously, when we analyzed ExPre-derived mCherry-alone and mCherry-eGFP puncta engulfed by glial cells in more detail, we found that 93% of mCherry-alone puncta were localized in Cathesin D-positive lysosomes in astrocytes whereas only 4.6% of mCherry-eGFP puncta were. Instead, the similar ratio of mCherry-alone and mCherry-eGFP puncta were found inside of Rab5-positive endosomes indicating that eGFP lost its signal during its transport from endosomes to lysosomes. These detailed data can be found in Extended data Figure 3 from our previous publication⁴.

14. The data in figure 8 and figure 9 assess brain atrophy and ventricle enlargement in both genotypes and models 14 days after the stroke. Were there any differences in the infarct volume 24h after the stroke between genotypes? A TTC staining, Nissl staining, or MRI imaging should be shown to prove that the initial infarct volume is the same across the different transgenic mice. The initial infarct volume could drastically influence the outcome and the development of the disease at later timepoints. Also, a quantification of the hemorrhagic area by DAB staining should be provided. There is an evident difference between WT and C-MEGF10KO and A-MEGF10KO hemorrhagic area that could be affecting the whole process in the acute timepoints.

Response: We agree with the reviewers' concern. Therefore, we performed additional experiments to determine whether there was difference in the initial brain injury between genotypes. Mice were sacrificed 24 hours after ischemic or hemorrhagic stroke. We examined brain infarct volume using Nissl staining. The representative images of brain coronal sections and quantification of infarct volume were shown in **Supp Fig. 5a-d**. Further, we examined the area of hemorrhagic region using DAB staining at 24 hours after hemorrhagic stroke (**Supp Fig. 5e-h**). The results demonstrated that there was no statistical difference of original brain infarct volume or hemorrhagic area between transgenic and control group of mice.

15. In figure 7 and 8, the authors show that MEGF10 and MERTK cKO mice have reduced brain atrophy. As both stroke models have been described to cause severe cellular damage and induce substantial cell death, does this reduction in brain atrophy purely reflect protection of synapses? Or do these genetic manipulations also affect cell death and/or axonal degeneration independent of protecting synapses?

Response: To examine if genetic manipulations affect cell death and axonal degeneration independent of protecting synapses in the acute stage of ischemic and hemorrhagic stroke, we quantified the number of TUNEL⁺ cells at 1 day post induction, and found there was no statistical difference between transgenic and control groups of mice, suggesting MEGF10 and MERTK genetic manipulation did not affect cell death after stroke (**Supp Fig. 9**). To study their impact on axonal degeneration, we used SMI32 and MBP colabeling to detect damaged myelinated axons. We found knockout of MEGF10 or MERTK decreased axonal degeneration in ischemic stroke mouse brain, but not in hemorrhagic stroke mouse brain, suggesting that in addition to protecting synapses, genetic manipulation of MEGF10 or MERTK also reduces axonal degeneration in ischemic stroke.

16. Are MEGF10 and MERTK detected in their single cell RNA seq analysis? Given the central role of both genes this data should be included into the figures 10 and S3.

Response: Thank you for your comments. MEGF10 and MERTK were detected in our single cell RNA seq analysis. The data was added to **Fig. 10c** and **Supp Fig. 6c**.

17. Sup Fig.4 shows the expression of MEGF10 and MERTK in hemorrhagic post mortem brain tissue; however, no control tissue was assessed. This data should be described in the result section and compared to controls to better interpret this result.

Response: As it is hard to get normal human brain tissue, the brain tissue proximal (500 μ m proximal to the hemorrhagic core) to the hemorrhagic area was considered healthy control tissue, and the expression of MEGF10/MERTK data was added to **Supp Fig. 8**.

18. For scRNA-seq experiments, the authors state that 12,000 total cells were analyzed per condition, how many astrocytes and microglia were analyzed? As the number of cells analyzed is the best predictor of the quality of scRNA-seq data sets, this information should be included for each cell type.

Response: Thank you for your comments. We updated the distribution of cell types in tSNE and used histogram to show the proportion of each cell type in each group (**Supp Fig. 10a**). Among them, the number of astrocytes in control group, hemorrhagic group and ischemic group

was 532, 434 and 414, respectively. The number of microglia/macrophages in control group, hemorrhagic group and ischemic group was 2247, 3941 and 6024, respectively (**Supp Table. 1**).

Supp Table. 1

sampleid	celltype_cell number						
	oligodendrocyte	Microglia/ Macrophage	T cell	neuroblast	endothelial cell	Astrocyte	Choroid plexus cells
control	3702	2247	33	748	1288	532	6
HS	2180	3941	1250	653	4	434	99
IS	3384	6024	1769	844	22	414	87

HS, hemorrhagic stroke; IS, ischemic stroke

19. In the scRNA-seq, were biological replicates pooled? Or were samples from each biological replicate run separately? If the latter, the authors should show the distribution of the cell types across biological samples to ensure equal distribution of the bio replicates on the tSNE plot.

Response: Thank you for your comments. For each group, we pooled the brain samples of 8 mice for sequencing.

20. For the scRNA-seq, why were no neuron or oligodendrocyte precursor cell clusters detected?

Response: In our scRNA-seq data we did not detect the clusters of neurons and oligodendrocyte precursor cells. One of the potential reasons is that different cells respond differently to enzymatic digestion⁸, and we have to acknowledge that the digestion method we used is not optimized for dissociating neurons and oligodendrocyte precursor cells from the adult mouse brain^{9, 10}. Neurons and oligodendrocyte precursor cells in the adult mouse brain are very sensitive and vulnerable to enzymatic digestion^{10, 11}, resulted in low number of neurons and oligodendrocyte precursor cells, which is below the detection threshold. Indeed, many researchers use single nuclear RNA-seq for neuron related work to overcome this shortcoming^{12,13}. As our current study mainly focus on comparing the function of astrocytes and microglia between ischemic and hemorrhagic stroke, we proceed to analyze our scRNA-seq data.

21. Given the work of Liddel et al. Nature 2017, it would be informative to compare the gene expression profiles of the astrocytes in the different ischemia models to the A1 vs. A2 astrocytes. If the astrocytes in the hemorrhagic stroke more closely resemble the A1 cells, this would be consistent with this previous study that showed that A1 astrocytes are less efficient at phagocytosing cellular material and could explain why there are differences in astrocyte engulfment between the two models.

Response: Thank you for your comments. According to the A1 and A2 related marker genes reported in previous studies^{14,15}, we built a database of A1 and A2 specific marker gene list. Based on the AddModuleScore function in Seurat, we evaluated and scored the A1 and A2 functions of astrocytes in each sample. As shown in **Supp Fig. 10b**, we found that astrocytes in the ischemic group had more A1 properties, while astrocytes in the hemorrhagic group were neither A1-like nor A2-like, suggesting that the differences in astrocyte engulfment between the two stroke models are not A1 or A2 phenotype dependent. Since the biological function of

astrocytes with different phenotypes is still under study, our study may provide new insight into phagocytic abilities of A1 and A2 astrocytes. We added this paragraph in the discussion section.

A1- and A2- astrocyte specific markers:

A1-specific	C3	A2-specific	Clcf1
	H2-T23		Tgm1
	Serping1		Ptx3
	H2-D1		S100a10
	Ggta1		Sphk1
	Ligp1		Cd109
	Gbp2		Tm4sf1
	Fbln5		B3gnt5
	Ugt1a		Ptgs2
	Fkbp5		Emp1
	Psmb8		Slc10ab
	Srgn		Cd14
	Amigo2		Stat3

Minor points:

1. *The graphical abstract does not summarize the findings of the paper well. A better representation should be selected.*

Response: Thanks for the suggestion. The graphical abstract was changed with improved illustration.

2. *The authors evaluate the neurological outcome in the ischemic stroke model using four different tests (mNSS, Rotarod test, Grid walking test, and Step-through passive avoidance test). It is unclear why in the hemorrhagic model the step-through passive avoidance test was not performed?*

Response: In our pilot study, we found hemorrhagic mice were not sensitive to step-through passive avoidance test, but sensitive to grid-walking test, and grid-walking test is routinely used to assess the behavior of hemorrhagic stroke mouse^{16,17}. Step-through passive avoidance test was used to assess cognitive and memory function which was closely related to hippocampus, while in our hemorrhagic model, injury was only induced in the striatum, but not hippocampus, that's why we performed grid-walking instead of step-through passive avoidance test in hemorrhagic mice.

3. *In line 166, the authors say that MEGF10 and MERTK were elevated 7 and 14 days after stroke. Figure 4d, however, does not show a significant increase at day 7.*

Response: The text was reworded. Thank you.

4. *In figure 5, the representation of the statistical significance is not clear. When comparing two different groups, the symbol should be different, or lines indicating which two groups are being compared should be added for clarity.*

Response: Thanks for the suggestion. We used different symbols and lines indicating different groups in the statistical graphs. Additionally, exact *p*-values are now directly included in the graphs.

5. In Figure 9c, the statistical analysis showed in the graphs does not match the one described in the text. Fig 9c is described in the text as "MEGF10KO in astrocytes showed no effects on vascular enlargement" however, in the figure, there is a # between MEGF10WT and A-MEGF10KO indicating a statistical difference of $p < 0.05$. This should be corrected.

Response: This has been corrected, thank you.

6. Figure 5j and 5l show that the number of engulfed puncta in microglia/macrophages is very similar to the number in astrocytes. However, the images shown in figure 1g and the text description suggest much more puncta are engulfed by microglia after hemorrhagic stroke compared to astrocytes.

Response: We reanalyzed engulfed synaptic puncta volume/size instead of the number in **Fig. 1** and **Fig. 5**. Quantification results indicated microglia/macrophages engulfed synaptic puncta 2-fold more than astrocytes in hemorrhagic stroke.

7. In Figures 1 and 4 authors only show merged images in what appear to be maximum intensity projections. Since it is difficult to appreciate the individual channels and co-localization, the authors should also show the individual channels and single imaging planes.

Response: The images in **Fig. 1** and **Fig. 4** have been replaced with single imaging planes including merged and individual channels, which show the co-localization in the same plane.

8. Line 107: Fig. 1e should be Fig. 1g. Same for line 114.

Response: This has been corrected, thank you.

9. Line 111: It describes Fig. 1c, but it is not specified in the text. It should be included for clarity.

Response: The manuscript text are reworded below: Line 112: "...and postsynaptic (Homer-1) proteins in the microgliosis and astrogliosis areas (**Fig. 1b**)"

10. The manuscript should be edited for grammar. There are numerous instances of grammatical errors throughout.

Response: We apologize for grammatical errors. We asked a professional editorial company (American Journal Experts) help to edit our revised manuscript. We believed that the manuscript is now ready for publication.

Reviewer #2 (Remarks to the Author):

This is an elegantly presented and comprehensive study of microglial and astrocytic involvement in synapse elimination in ischemic and haemorrhagic stroke. Synapse engulfment and phagocytosis by these cells have been shown recently, however, the comparison between

the two pathological events using advanced tools and showing the beneficial effects of blocking phagocytosis provide advances in the field. In particular, it is interesting that astrocytes seem to have less direct contribution to synapse phagocytosis in haemorrhagic stroke. The systematic approach using immunohistochemical, biochemical, transcriptional and behavioural analyses and the tools developed by the authors, including synapse reporters, and various conditional knock-out mice have resulted in confidence in the findings. I have relatively minor comments, which should be addressed by the authors.

Minor comments:

1. The EM images in Fig. 6 are low resolution and the blue overlays are obscuring the demonstration of synaptic inclusions (SI) in the cells. Some of these would need replacement with higher quality/higher resolution images in which the synaptic vesicle can be clearly visualised in the SI.

Response: The obscure images were replaced with higher quality/higher resolution images in the revised **Fig. 6**.

2. The authors should specify in the methods if only one-way ANOVA or two-way ANOVA tests were also used in the study. In addition, the type of statistical tests used should be indicated in the figure legends.

Response: A detailed description of the statistical method was given in the method section of the revised manuscript, and the types of statistical tests were indicated in the figure legends.

3. It should also be specified in the methods if the scRNA-seq analysis was performed in one batch or in several batches. If the latter, was any computational batch correction used? They should also comment on whether any filtering was carried out for cells with disproportionately high mitochondrial genes (possibly dead cells).

Response: To better illustrate the way we perform scRNA-seq analysis, we added the table below in the **Supp Table 1**. To remove the batch effects in single-cell RNA-sequencing data, the mutual nearest neighbors (MNN) presented by Haghverdi et al¹⁸ was performed with the R package batchelor.

Supp Table 2.

sampleid	specie	group	batchid
HS	mm10	HS	2
IS	mm10	IS	2
control	mm10	control	1

HS, hemorrhagic stroke; IS, ischemic stroke; mm, Mus musculus

To remove low quality cells and likely multiple captures, which is a major concern in microdroplet-based experiments, we applied a criterion to filter out cells with UMI/gene numbers out of the limit of mean value +/- 2-fold of standard deviations assuming a Gaussian distribution of each cells' UMI/gene numbers. Following visual inspection of the distribution of cells by the fraction of mitochondrial genes expressed, we further discarded low-quality cells where >30% of the counts belonged to mitochondrial genes. The number of cells before and after filtration is as follows (**Supp Table 2, Supp Fig. 10c**) :

Supp Table 3.

sampleid	Mean_nUMI_	Mean_nGene_	Mean_mito.percent_	Total_cells_	Mean_nUMI_	Mean_nGene_	Mean_mito.percent_
	beforeQC	beforeQC	beforeQC	beforeQC	afterQC	afterQC	afterQC
control	6511.61	2068.80	0.092858	10638	6874.50	2247.01	0.068830
HS	7745.72	1962.84	0.063106	10140	8259.35	2119.52	0.048955
IS	7774.83	1955.81	0.059947	15190	8124.77	2094.42	0.045606

HS, hemorrhagic stroke; IS, ischemic stroke; QC, quality control

4. A heatmap for unbiased clustering and also for cell-type specific DEGS should be included, representing the cell-type identities and their marker genes. This is to better demonstrate how cell populations were identified in the datasets, and whether this was consistent between the two experimental groups.

Response: To better illustrate the cell-type identities and their marker genes, we added a dot graph instead of heatmap in **Supp Fig. 10d**, which was consistent between ischemic and hemorrhagic groups.

5. It is not indicated in any of the scRNA-seq-related figures whether the microglial and astrocytic expression of *MEGF10* and *MERTK* was altered in the dataset. If this was not the case, then it should be discussed how this relates to the protein level findings (by WB) in the main part of the manuscript.

Response: Thank you for your comments. *MEGF10* and *MERTK* were detected in our single cell RNA seq analysis (**Fig. 10c** and **Supp Fig. 6c**). Interestingly, we did not observe significant difference in the expression of *MEGF10* and *MERTK* among the three groups, which are inconsistent with our western blot results. The possible reason is that single cell RNA sequencing is a genomic approach for the detection and quantitative analysis of messenger RNA molecules in each cell, while western blot is a method used to detect specific protein from a mixture of proteins. The expression of mRNA and proteins are time-dependent, and they may have different expression profiles at the same time. This point was added to the discussion section.

6. The difference in astrocytic phenotype/function between the ischemic and hemorrhagic stroke is interesting. The potential causes of this should be also discussed more explicitly in the manuscript. In specific, it would be pertinent to include if there is any evidence in their scRNA-seq data for a different inflammatory/immune cell composition in the two lesion types, which may have had a differential impact on astrocytes.

Response: As shown in Table 4, the proportion and number of T cells in the hemorrhagic group was 14.60% and 1250, respectively, and those in the ischemic group was 14.10% and 1769, respectively. Considering that the composition of T cells in these two experimental groups was similar, we assume inflammatory/immune cells may have a negligible impact on astrocytes.

Supp Table 1.

sampleid	celltype_cell number						
	oligodendrocyte	Microglia/ Macrophage	T cell	neuroblast	endothelial cell	Astrocyte	Choroid plexus cells
control	3702	2247	33	748	1288	532	6
HS	2180	3941	1250	653	4	434	99

IS	3384	6024	1769	844	22	414	87
----	------	------	------	-----	----	-----	----

HS, hemorrhagic stroke; IS, ischemic stroke

Reviewer #3 (Remarks to the Author):

The interesting study by Shi et al. focuses on the role of reactive gliosis in stroke and in cellular and functional recovery in ischemic and hemorrhagic stroke. The authors report some exciting findings regarding 1) molecular and functional differences in microglia and astrocytes in the glial scar in these two different types of strokes, and 2) the distinct roles that these cell types play in the extent of stroke-induced injury, and in regeneration and functional recovery. The authors specifically delete MEGF10 and MERTK phagocytic receptors in microglia and astrocytes to inhibit phagocytosis in these distinct cell types. In hemorrhagic stroke, inhibiting phagocytosis in microglia/macrophages, but not in astrocytes, improved cellular, anatomical and functional outcomes. Conversely, in ischemic stroke, inhibiting phagocytosis in microglia/macrophages and in astrocytes, improved brain damage and functional outcomes. The authors conclude that reactive microgliosis and astrogliosis play individual roles in mediating synapse engulfment in different stroke models, and preventing the individual roles of these cells can rescue synapse loss and differentially attenuate the consequences of stroke.

This is an important and novel report, as it demonstrates not only differences in reactive gliosis in different stroke models, but also the impact of phagocytosis in distinct cell types during the recovery process. These findings could eventually lead to designing more targeted therapies in different types of stroke aimed at attenuating brain injury and improving functional recovery.

I would recommend some revisions, before considering this article for publication.

1. Figure 4. Western blot in Panel d. the levels of MEGF10 dramatically decrease at D3, between D1 and D7. However, the quantification does not show that. The Western blot should match the quantification and vice-versa.

Response: We reformed western blot and changed the representative blot images with improved images in the revised **Fig. 4a, k**.

2. Figure 6. This figure shows that conditional MEGF10 and MERTK deletion reduces glial-cell mediated synapse engulfment after stroke, and that there are differential effects in hemorrhagic vs. ischemic stroke. However, this figure is not quantitative, and quantification of engulfed synaptic elements should be performed to convincingly support the morphological data.

Response: In revised **Fig. 6**, we measured the number of synaptic elements within glial cells of each group, and found A-MEGF10^{KO}, C-MEGF10^{KO}, A-MERTK^{KO}, C-MERTK^{KO} mice engulfed much less synaptic elements than control group after ischemic stroke, while C-MEGF10^{KO} and C-MERTK^{KO} mice engulfed less synaptic elements than control group after hemorrhagic stroke.

3. Page 9. Figure 7. In hemorrhagic stroke, MEGF10 and MERTK deletion only slightly increased the number of total dendritic spines and mature spines in the gliosis region, as shown in Figure 7h-l. The difference is much larger in ischemic stroke. This should be reworded in the text.

Response: The text has been reworded as follows: Line 280: "...significant increase in the number of total spines (2.2-fold and 3.0-fold, respectively), mature spines (1.9-fold and 2.5-fold, respectively) and filopodia spines (3.5-fold and 2.9-fold, respectively), as compared to the control (Fig. 7c-g)." Line 287: "...total dendritic spines and mature spines in the gliosis region (\approx 1.5-fold change, Fig. 7h-l)..."

4. The authors convincingly demonstrate differences in brain atrophy in distinct types of stroke after MEGF10 and MERTK deletion. It would be important to also characterize changes in white matter, and how these changes affect brain atrophy in different KO mouse lines in distinct stroke models.

Response: To study their impact on axonal degeneration, we used SMI32 and MBP co-labeling to detect damaged myelinated axons. We found knockout of MEGF10 or MERTK decreased axonal degeneration in ischemic stroke mouse brain, but not in hemorrhagic stroke mouse brain (**Supp Fig. 9**), suggesting that in addition to protecting synapses, genetic manipulation of MEGF10 or MERTK also reduces axonal degeneration in ischemic stroke.

5. Figures 8 and 9. I am not sure about the rationale for organizing these two figures in the current layout. I would recommend a different layout by combining brain atrophy/ventricular enlargement data in one figure (Figure 8) and all behavioral data in a separate figure (Figure 9).

Response: To make the layout clearer, we followed the reviewer's suggestion. We combined the brain atrophy and ventricular enlargement data in revised **Fig. 8**, and also combined all the behavioral data in revised **Fig. 9**.

6. The grammar and English of the whole manuscript need significant revisions.

Response: The grammar and English in the manuscript have been revised by Academic editing service company named American Journal Experts.

References:

1. Alawieh A, Langley EF, Tomlinson S. Targeted complement inhibition salvages stressed neurons and inhibits neuroinflammation after stroke in mice. *Sci Transl Med* **10**, (2018).
2. Alawieh AM, Langley EF, Feng W, Spiotta AM, Tomlinson S. Complement-Dependent Synaptic Uptake and Cognitive Decline after Stroke and Reperfusion Therapy. *J Neurosci* **40**, 4042-4058 (2020).
3. Chung WS, *et al.* Astrocytes mediate synapse elimination through MEGF10 and MERTK pathways. *Nature* **504**, 394-400 (2013).
4. Lee JH, *et al.* Astrocytes phagocytose adult hippocampal synapses for circuit homeostasis. *Nature*

- 590, 612-617 (2021).
5. Hammond TR, *et al.* Single-Cell RNA Sequencing of Microglia throughout the Mouse Lifespan and in the Injured Brain Reveals Complex Cell-State Changes. *Immunity* **50**, 253-271 e256 (2019).
 6. Zhu Y, *et al.* Identification of different macrophage subpopulations with distinct activities in a mouse model of oxygen-induced retinopathy. *Int J Mol Med* **40**, 281-292 (2017).
 7. Gilles JF, Dos Santos M, Boudier T, Bolte S, Heck N. DiAna, an ImageJ tool for object-based 3D co-localization and distance analysis. *Methods* **115**, 55-64 (2017).
 8. Marsh SE, *et al.*, (2020).
 9. Chen R, Wu X, Jiang L, Zhang Y. Single-Cell RNA-Seq Reveals Hypothalamic Cell Diversity. *Cell Rep* **18**, 3227-3241 (2017).
 10. Marques S, *et al.* Oligodendrocyte heterogeneity in the mouse juvenile and adult central nervous system. *Science* **352**, 1326-1329 (2016).
 11. Leng K, *et al.* Molecular characterization of selectively vulnerable neurons in Alzheimer's disease. *Nat Neurosci* **24**, 276-287 (2021).
 12. Lacar B, *et al.* Nuclear RNA-seq of single neurons reveals molecular signatures of activation. *Nat Commun* **7**, 11022 (2016).
 13. Alkaslasi MR, *et al.* Single nucleus RNA-sequencing defines unexpected diversity of cholinergic neuron types in the adult mouse spinal cord. *Nat Commun* **12**, 2471 (2021).
 14. Liddelow SA, *et al.* Neurotoxic reactive astrocytes are induced by activated microglia. *Nature* **541**, 481-487 (2017).
 15. Clarke LE, Liddelow SA, Chakraborty C, Munch AE, Heiman M, Barres BA. Normal aging induces A1-like astrocyte reactivity. *Proc Natl Acad Sci U S A* **115**, E1896-E1905 (2018).
 16. Sun Y, *et al.* Neuroprotection and sensorimotor functional improvement by curcumin after intracerebral hemorrhage in mice. *J Neurotrauma* **28**, 2513-2521 (2011).
 17. Barratt HE, Lanman TA, Carmichael ST. Mouse intracerebral hemorrhage models produce different degrees of initial and delayed damage, axonal sprouting, and recovery. *J Cereb Blood Flow Metab* **34**, 1463-1471 (2014).
 18. Haghverdi L, Lun ATL, Morgan MD, Marioni JC. Batch effects in single-cell RNA-sequencing data are corrected by matching mutual nearest neighbors. *Nat Biotechnol* **36**, 421-427 (2018).

Reviewers' Comments:

Reviewer #1:

Remarks to the Author:

The authors have done an excellent job revising the manuscript. There are only a few very minor points that remain to be addressed:

1. The authors used the number of slices for statistical comparison when they should have used the number of analyzed mice in Figs. 1-7 and S4d-g, S7, and S9 (e.g., $n=13-16$ in Fig. 1f, which should be $n=4$ because only 4 mice were analyzed per group).
2. The organization of the Figures is often different from the order results are presented in the text.
3. In Supp Fig. 5 they say that they do a DAB staining but in the legend, it states a Nissl staining (g-h).
4. In Supp Fig.1 they represent the differential contribution of microglia/macrophages as % of those cells that are engulfing synapses. This may not be the best way to evaluate the contribution of each cell type. The total number of cells engulfing would be a beneficial addition. For example, if the % of engulfment is the same across macs and microglia, but there are more macrophages present compared to microglia, the contribution of macrophages to the engulfing process as a whole may be higher.

Reviewer #2:

Remarks to the Author:

The revision has substantially improved the manuscript and all my concerns raised in my first review had been fully addressed. I now recommend the manuscript for publication.

Reviewer #3:

Remarks to the Author:

The authors have comprehensively addressed all the issues raised by myself and the other reviewers. They have extensively revised the manuscript and almost all figures, and added a significant body of new data and revised analysis of previous data. The manuscript has significantly improved and in my opinion continues to be a very important and exciting report.

Response to comments of the reviewers

Reviewer #1 (Remarks to the Author):

The authors have done an excellent job revising the manuscript. There are only a few very minor points that remain to be addressed:

1. The authors used the number of slices for statistical comparison when they should have used the number of analyzed mice in Figs. 1-7 and S4d-g, S7, and S9 (e.g., n=13-16 in Fig. 1f, which should be n=4 because only 4 mice were analyzed per group).

Response: Thanks for the suggestions. We reworded the n number as 'n= X mice' and provided exact sample size used to achieve statistics of relevant figures in the revised manuscript.

2. The organization of the Figures is often different from the order results are presented in the text.

Response: Thanks for the suggestions. There are many panels in each figure. We have re-organized and re-worded the results and tried our best to describe the results as the order of figures presented in the text in the revised manuscript.

3. In Supp Fig. 5 they say that they do a DAB staining but in the legend, it states a Nissl staining (g-h).

Response: The figure legend was reworded in the revised **Supp Fig. 6**.

4. In Supp Fig. 1 they represent the differential contribution of microglia/macrophages as % of those cells that are engulfing synapses. This may not be the best way to evaluate the contribution of each cell type. The total number of cells engulfing would be a beneficial addition. For example, if the % of engulfment is the same across macs and microglia, but there are more macrophages present compared to microglia, the contribution of macrophages to the engulfing process as a whole may be higher.

Response: We followed the reviewer's suggestion. We quantified the number of phagocytic microglia (SYP⁺/P2RY12⁺ and Homer1⁺/P2RY12⁺ cells) and macrophage (SYP⁺/F4/80⁺ and Homer1⁺/F4/80⁺ cells) and found that microglia contributed more than macrophage in both ischemic stroke and hemorrhagic stroke (**Supp Fig. 2**).

Reviewer #2 (Remarks to the Author):

The revision has substantially improved the manuscript and all my concerns raised in my first review had been fully addressed. I now recommend the manuscript for publication.

Reviewer #3 (Remarks to the Author):

The authors have comprehensively addressed all the issues raised by myself and the other reviewers. They have extensively revised the manuscript and almost all figures, and added a significant body of new data and revised analysis of previous data. The manuscript has significantly improved and in my opinion continues to be a very important and exciting report.